# Improving the representativeness of UK's national COVID-19 Infection Survey through spatio-temporal regression and post-stratification

Koen B. Pouwels [1,2] ✉, David W. Eyre [2,3,4,5], Thomas House [6,7], Ben Aspey[8], Philippa C. Matthews[9,10,11], Nicole Stoesser [2,4,5,10], John N. Newton [12], Ian Diamond[8], Ruth Studley[8], Nick G. H. Taylor[8], John I. Bell[13], Jeremy Farrar[14], Jaison Kolenchery[4,10], Brian D. Marsden [10,15], Sarah Hoosdally[10], E. Yvonne Jones [10], David I. Stuart[10], Derrick W. Crook[2,4,5,10], Tim E. A. Peto[2,4,5,10], A. Sarah Walker[2,10,16] & the COVID−19 Infection Survey Team*

Population-representative estimates of SARS-CoV-2 infection prevalence and antibody levels in specific geographic areas at different time points are needed to optimise policy responses. However, even population-wide surveys are potentially impacted by biases arising from differences in participation rates across key groups. Here, we used spatio-temporal regression and post-stratification models to UK's national COVID-19 Infection Survey (CIS) to obtain representative estimates of PCR positivity (6,496,052 tests) and antibody prevalence (1,941,333 tests) for different regions, ages and ethnicities (7-December-2020 to 4-May-2022). Not accounting for vaccination status through post-stratification led to small underestimation of PCR positivity, but more substantial overestimations of antibody levels in the population (up to 21 percentage points), particularly in groups with low vaccine uptake in the general population. There was marked variation in the relative contribution of different areas and age-groups to each wave. Future analyses of infectious disease surveys should take into account major drivers of outcomes of interest that may also influence participation, with vaccination being an important factor to consider.

The Covid-19 pandemic has a devastating impact on morbidity, mortality and economies around the world. As of 30 September 2022, there have been over 600 million confirmed COVID-19 cases, including over 6.5 million deaths according to the World Health Organization[1]. These numbers substantially underestimate the true number of cases due to the lack of systematic testing in most countries. The United Kingdom (UK) has been a noticeable exception in terms of SARS-CoV-2 surveillance, recognising early on the value of investment in large population-based studies that follow a random sample of the population longitudinally with testing performed at fixed intervals independent of symptoms. This approach provides much more reliable estimates of levels and trajectories of the prevalence of SARS-CoV-2 infections and antibody levels than solely having to rely on national testing systems[2]. In most countries, only people with specific

A full list of affiliations appears at the end of the paper. *A list of authors and their affiliations appears at the end of the paper.
✉e-mail: koen.pouwels@ndph.ox.ac.uk

symptoms, or those with contacts with known cases, are eligible for testing in systems set up by governments. However, a substantial proportion of individuals do not report any symptoms around their positive SARS-CoV-2 PCR test[3], and testing capacity, testing strategies and the probability that a symptomatic individual decides to get tested varies by time, socio-demographic factors and location[2,4]. This complicates the interpretation of such data sources that are not designed to provide representative estimates of the prevalence or incidence of SARS-CoV-2-positive individuals. To inform decisions around implementation or (dis)continuation of (local) mitigation measures, policymakers ideally would have population-representative estimates of how many people are infected with SARS-CoV-2 in small areas at different time points. Similarly, it is important to track how SARS-CoV-2 antibody levels change over time to enable the likely levels of protection at the national and more granular regional levels to be accounted for when making vaccination and other mitigation policy decisions.

We used data from the UK's national COVID-19 Infection Survey (CIS) to demonstrate how a spatio-temporal regression and post-stratification modelling approach, extending previously developed spatial regression and post-stratification using a time and space-time component[5], can be used to obtain representative temporal estimates of the swab positivity and antibody prevalence at the national and sub-regional level, and for different ages and ethnicities[2,5–7]. Early in the pandemic, the Office for National Statistics (ONS) and government representatives from England, Northern Ireland, Scotland and Wales divided the UK in 133 sub-regional areas that were deemed relevant for local policy-making and simultaneously sufficiently large to provide meaningful estimates of swab positivity and antibody prevalence. For the current analysis, we focused on the 116 areas in England given the availability of detailed administrative data on vaccination uptake during the study period.

The UK's CIS depends on voluntary participation. Therefore, despite invitations being sent to randomly selected addresses and monetary compensation for participation, it is possible that it is not optimally representative of the whole population. Here, we explore to what extent accounting for vaccination status in the sample compared to the general population improved estimates of swab and antibody positivity, recognising the possibility that individuals who are more likely to get vaccinated may also be more likely to participate in infectious disease surveys upon invitation.

In addition, given the large interest of policymakers in understanding why certain areas in England consistently had higher number of people testing positive in the national testing programme, we evaluated whether this observation might be explained by regional variation in the probability of deciding to take a test upon symptoms, or whether similar trends were observed in the CIS where survey participants are tested based on a fixed schedule independent of symptoms status. Furthermore, given the fact that regional variation in a long list of health outcomes and behaviours can be explained by deprivation and urbanicity[8,9], we evaluated to what extent areas that frequently have higher swab positivity estimates compared to other areas are more deprived and more urban.

## Results

Between Monday 7 December 2020 and Wednesday 4 May 2022, 6,496,052 PCR test results, taken following an external assessment schedule without knowledge of symptom status from participants in England, were available for analysis. Of these tests, 120,436 (1.9%) were positive. During the same period, 1,941,333 blood samples from participants in England were tested for SARS-CoV-2 anti-spike IgG antibody levels, of which 1,360,860 tests were above the 100 binding antibody units (BAU) per millilitre thresholds. This threshold corresponds to the antibody level estimated to confer 67% protection against Delta infection[10].

### Swab positivity

There was marked variation in PCR positivity over time, with the estimated prevalence ranging from 0.09% in week 18 of 2021 to 7.16% in week 12 of 2022 in England. While the South-West region had lower PCR positivity for most of the study period, its Omicron BA.2 peak was more pronounced than in the London region that generally experienced higher prevalence of most variant waves, with a particularly high Omicron BA.1 peak (Fig. 1 and Supplementary Data 1). Similarly, while individuals of black ethnicity experienced a large BA.1 peak compared to other ethnicities, they experienced a less pronounced BA.2 peak than other ethnicities (Fig. S1). PCR positivity varied markedly between waves for different age groups (Fig. S2). For example, adolescents aged 12–15 had by far the highest PCR positivity peak during the Delta wave, while their PCR positivity rates were consistently lower than for other age categories during the Omicron BA.2 wave.

Study participants were more likely to be vaccinated than the overall population (e.g., 93% in the survey vs 75% based on the admin data by May 2022, Fig. 2). The best fitting spatiotemporal multilevel regression model for PCR positivity, based on the Watanabe-Akaike information criterion (WAIC), was a model that included terms for age, sex, ethnicity, vaccination status, a Besag-York-Mollié (BYM2) specification for the CIS area effect, and two-way interactions between time – measured in weeks – and age, ethnicity, vaccination status, and CIS area. The only covariate term that did not lead to a clear improvement in the WAIC was the interaction between sex and time. The inclusion of the latter term resulted in an increase instead of the decrease of WAIC of ≥ 20 observed for other covariates. The interaction between CIS area and time was modelled using a so-called type IV space-time interaction, assuming that for the ith area, the parameter vector has a temporal dependency structure on the time component and that at each time point, there is a spatial correlation. The combination of a type IV space-time interaction, whereby time was modelled using a first-order random walk was chosen based on the WAIC, having a better model fit than other interaction types and/or other ways to model time (WAIC difference ≥ 30).

While survey participants were more likely to be vaccinated than expected based on the national administrative data on vaccination uptake, adding an indicator for vaccination status (yes/no) interacting with time in weeks to a model that already accounted for all other variables improved model fit (WAIC difference of 964), but this had only small effects on post-stratified estimated levels of PCR positivity (Fig. 1). When not accounting for over-representation of vaccinated individuals in the survey, the largest underestimation of positivity across England overall was 0.35 percentage points (pp) (6.22% vs 6.57%). The underestimation of swab positivity was more pronounced for crude swab positivity based on the raw data during most of the study period.

When focusing on subregional estimates of swab positivity, dividing the nine regions of England into 116 sub-regions (CIS areas), because of low positivity rates, many areas had weeks with 2 or fewer positive samples in total despite the size of the survey meaning several thousand were tested (25th percentile of the number of positive tests per week was 2), emphasising the need for small-area estimation and potentially explaining why a better model fit was obtained for models with a type IV interaction and a random walk structure for time. When focusing on the impact of post-stratifying for vaccination among these smaller areas, the largest differences were observed for areas within London, with Lambeth having the most number of weeks (3 out of 74 weeks) with a difference between point estimates that was larger than 1pp. The unadjusted positivity among survey participants led to larger underestimations of PCR positivity until the Omicron BA2 peak when there were shifts in terms of ethnic and age-related risk groups (Fig. 1 and Fig. S1, Fig. S2).

Post-stratified estimates of the fully adjusted model including 95% credible intervals (CrI) aggregated by England, region, CIS area, age,

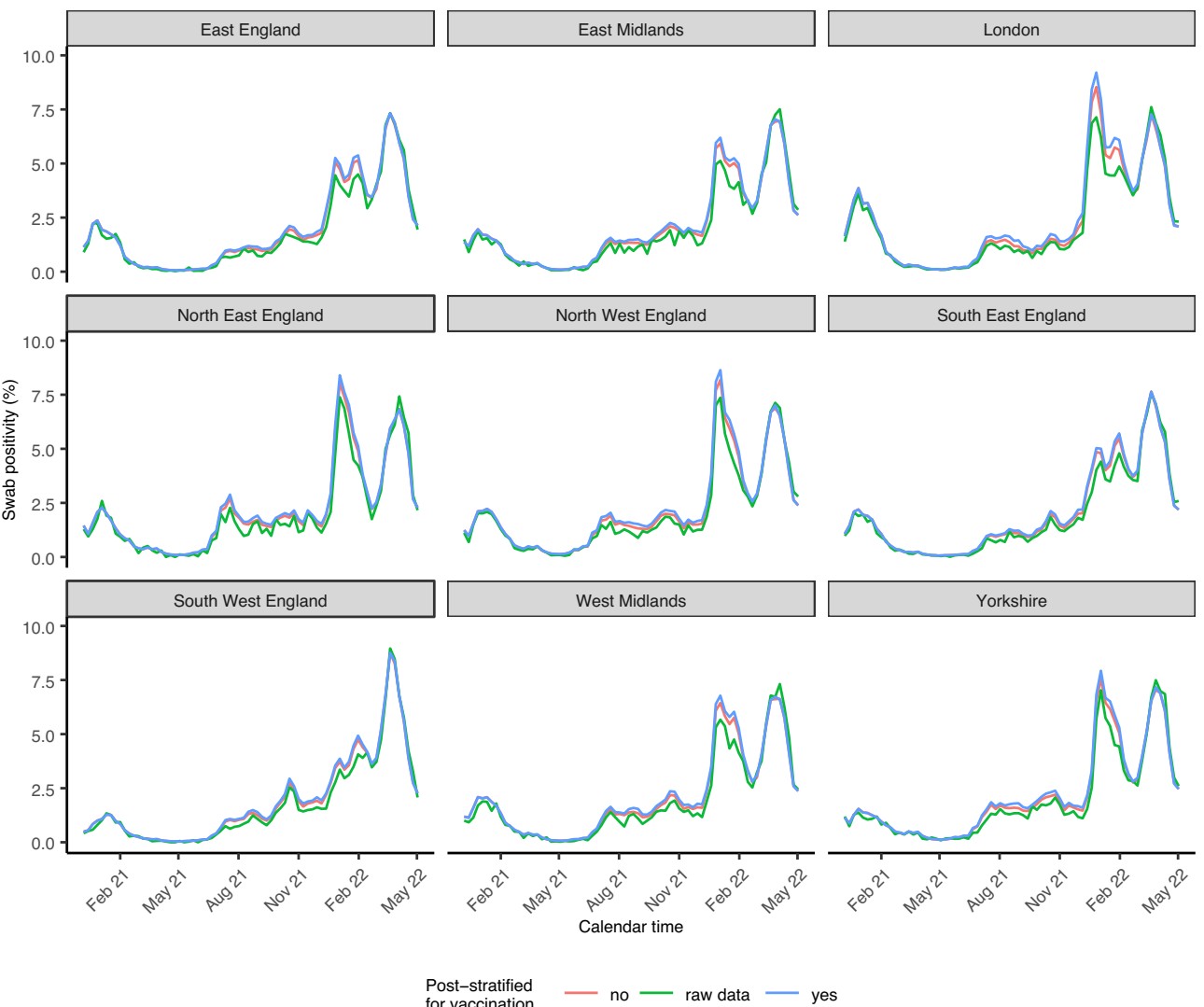

**Fig. 1 | Post-stratified modelled estimates of swab PCR positivity by region over time.** Modelled estimates are post-stratified for age, sex, CIS area (116 sub-regions within the 9 administrative regions shown), ethnicity and vaccination status. Estimates are presented as posterior medians (solid lines) with shading representing a 95% credible interval. Crude numbers from the underlying data are labelled as raw data. Vertical black lines indicate from left to right the start (at the national level) of the Alpha, Delta, Omicron BA1, and Omicron BA2 dominant period, respectively.

and ethnicity are provided in Supplementary Data 1. While CrI width varies over time and depends on the PCR positivity, at the smallest geographic CIS level the average 95% CrI width was 1.08 pp.

There was large variability over time in which areas of England had the highest PCR positivity, but some areas also consistently had higher or lower PCR positivity (Fig. 3 and Fig S3). Using the spatiotemporal regression and post-stratification model that accounted for vaccination status, we evaluated whether certain areas consistently had a high probability (≥ 80%) of being ranked among the top 10 areas in terms of the highest weekly PCR positivity. Two areas out of a total of 116 areas (1.7%) – Kirklees and Rochdale – had a high probability of being ranked in the top 10 areas of highest swab positivity over more than 25% of the study period (>18 out of 74 weeks) (Table 1), while 48/116 (41%) areas were never ranked in the top 10. During the times that the Kirklees and Rochdale areas had a high probability of being ranked in the top 10 areas of highest swab positivity, their posterior mean estimates were on average, respectively, 2 and 2.5 times higher than the average of all 116 CIS areas during those weeks. A linear regression with area-specific levels of deprivation (0.27, 95% CI 0.18-0.35 decrease per 1 unit increase in deprivation ranking; t statistic -6.14) and the percentage of the area considered rural (0.42, 95% CI 0.30-0.54 decrease per 1%

increase rurality; t statistic -6.97) as the only covariates explained 57% of the variance in median PCR positivity ranking of the areas, indicating that less deprived areas and more rural areas were likely to have lower PCR positivity compared to more deprived and urban areas. Although deprivation and urbanicity explained a large proportion of the variance in median PCR positivity ranking, including those covariates as contextual variables in the spatiotemporal MRP model in a sensitivity analysis did not improve the model fit of the main analysis model that captured contextual information only through the effects of area, time and the type-IV interaction.

## Antibody prevalence

We primarily estimated antibody prevalence at a threshold previously estimated to be associated with 67% protection against new infection with the Delta variant (100 BAU/ml)[10]. As observed when focusing on swab positivity, the best fitting model was a model with a type IV space-time interaction and first-order random walk for time modelled in weeks (WAIC difference of ≥ 29). In contrast to the MRP model for swabs, an interaction between sex and time improved the model fit, meaning that all considered covariate terms were included in the final model. Antibody positivity during the first week of the vaccination

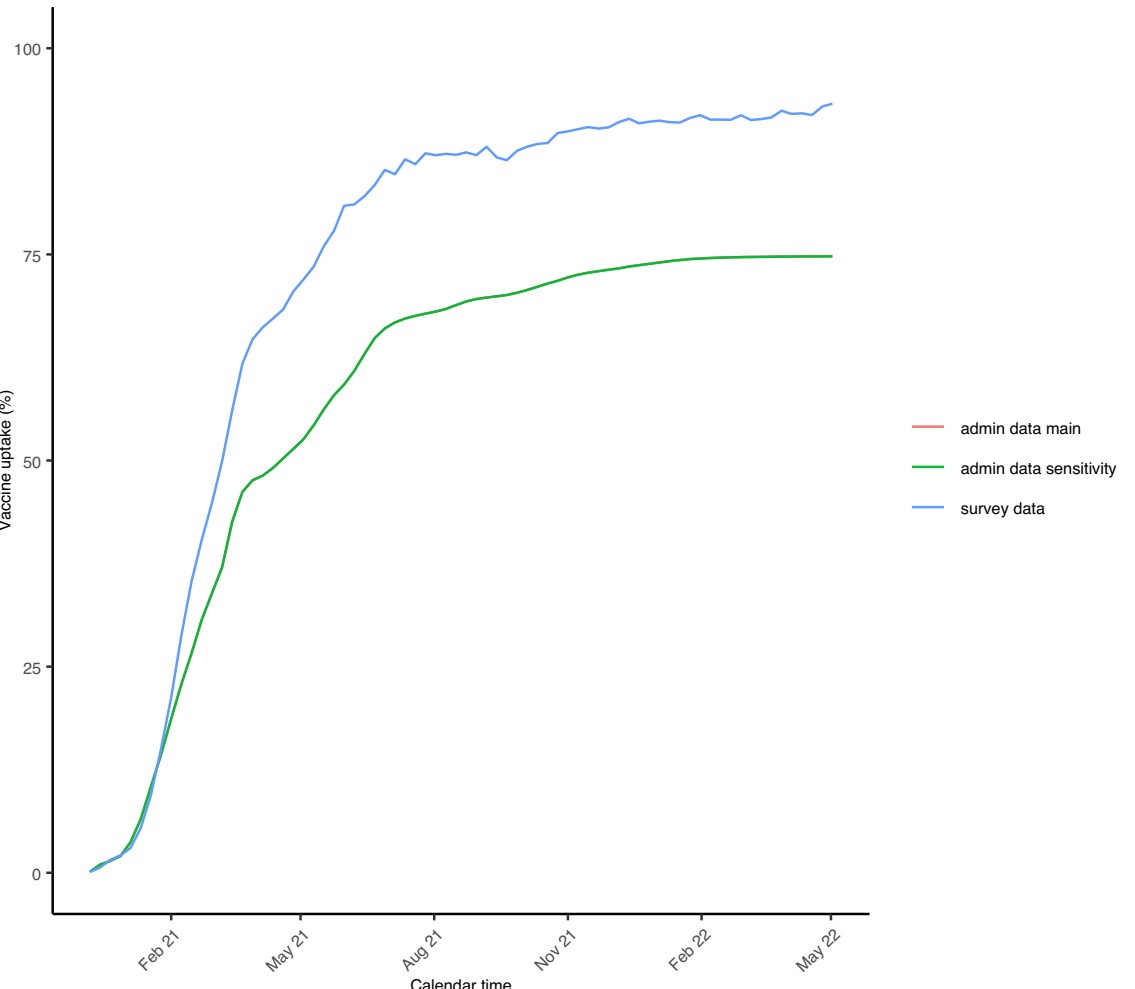

**Fig. 2 | Crude cumulative vaccination uptake based on survey participants and admin data.** The red and green lines almost perfectly overlap reflecting the fact that the assumptions we made about vaccination uptake among children too young to be present in the 2011 Census are of negligible importance given that during the study period very few children aged 9–11 years of age – the ages from which we extrapolated to younger children assuming I) no vaccinations under the age of 5 and half the coverage of that observed among 9–11 years old children for those aged 5–8 years of age (admin data main), or II) no vaccinations under the age of 9 (admin data sensitivity).

campaign in England, which started on 8 December 2020, was only 5.55% (95% CI 4.83-6.38%). The percentage of individuals having antibodies levels ≥ 100 BAU/ml increased over time, with steeper increases coinciding with increases in second and third vaccinations (Fig. 4).

Post-stratifying for vaccination status had negligible effects at the start of the vaccination campaign, but over time, when an increasing proportion of individuals had antibody levels above 100 BAU/ml due to vaccination rather than infection, the effect of accounting for over-representation of vaccinated individuals in the survey became more marked (Fig. 4). Not accounting for vaccination status led to point estimates for antibody positivity (≥ 100 BAU/ml) that were on average 5.2 pp higher than when taking this into account, with the largest difference (21 pp; 76% vs 55%) observed in Newham (London region) during the third week of July 2021. Similarly, using crude positivity estimates generally resulted in an overestimation during periods that the MRP model that did not account for vaccination status led to overestimates of antibody positivity (Fig. 4).

When using the spatiotemporal models to post-stratify and summarise positivity estimates by different population characteristics, differences between the models with and without vaccination status were most marked in areas and population groups with lower vaccine uptake according to the administrative data, e.g., for non-white ethnicities with the largest differences observed for black ethnicity (Fig.

S4), and certain age-categories with the largest differences in estimates occurring among those aged 25–34 years old (Fig. S4).

## Discussion

Using data from one of the largest SARS-CoV-2 community surveillance studies in the world that randomly invites individuals from private households to obtain representative estimates of SARS-CoV-2 PCR positivity and antibody levels, we found that those who agree to participate in the survey are more likely to be vaccinated than the overall population. Not accounting for vaccination status, as done throughout the pandemic, resulted in a small underestimation of PCR positivity but a more substantial overestimation of the percentage of the population having antibody levels at least as high as the threshold previously estimated to be associated with 67% protection against infection with the Delta variant[10]. While estimates were, as expected, similar with and without accounting for vaccination status at the start of the vaccination campaign, and 5.2 pp when taking the average difference for England over the entire period, the maximum difference in antibody prevalence at the aforementioned antibody threshold was 21 pp. Such large differences, observed in an area with one of the lowest percentages of White British populations and one of the highest poverty rates in England, could lead to underinvestment in interventions that could help increase the percentage of the local population with

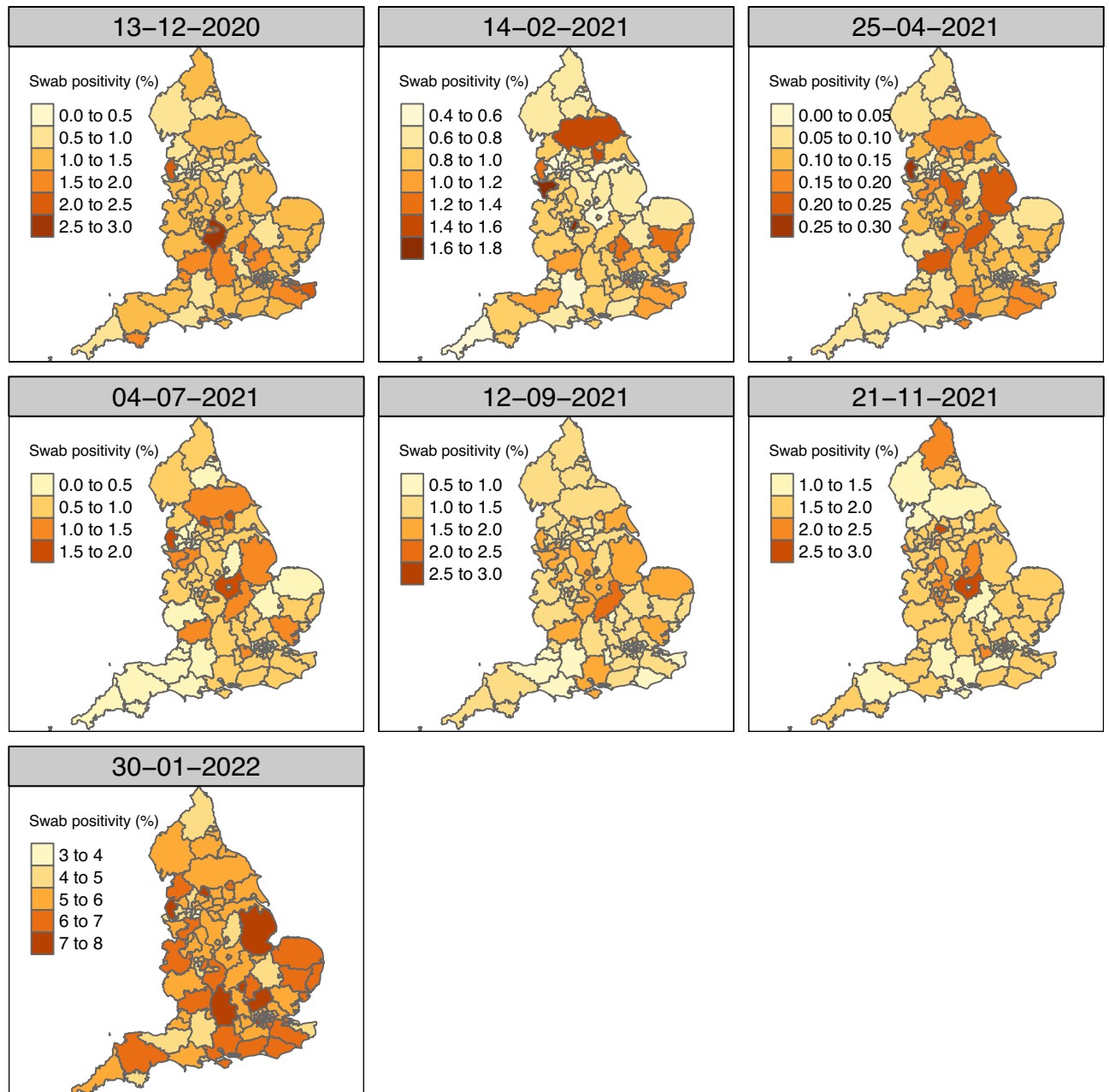

**Fig. 3 | Post-stratified estimate of swab PCR positivity by CIS area over time.** Estimates are post-stratified for age, sex, ethnicity and vaccination status.

sufficiently high antibody levels to be protected against new SARS-CoV-2 infections, potentially contributing to severe acute disease or long COVID-19 in the population.

Importantly, these findings suggest that when performing infectious disease surveillance for pathogens for which vaccines are available and commonly used, accounting for vaccination status should be recommended, as despite randomly inviting individuals and compensating them for participation, those that decide to participate in the survey may be more likely to be vaccinated than those that decide to decline the invitation. Accounting for vaccination status through post-stratification could have a particularly large impact among groups of the population with lower vaccination rates, as here also observed for non-white ethnicities, younger age groups, and those living in urban and more deprived areas[11]. These findings support the suggestion that even large studies that aim to randomly select individuals from the target population have to carefully consider how to prevent and minimise selection bias and account for vaccination status

when using carefully designed surveys for infectious diseases[12]. For example, the Census Household Pulse conducted by the US Census Bureau and eleven statistical government partners, substantially overestimated uptake of the first dose vaccination by 14 pp in May 2021 despite accounting for age, gender, education, and ethnicity[12]. In comparison, on 23 May 2021, crude estimates of uptake of the first dose vaccination were approximately 20 pp higher in the CIS compared to estimates based on the administrative data.

Areas with higher levels of deprivation and more urban areas more often had a higher PCR positivity than other areas, explaining a large percentage of the observed variation in the median ranking of PCR positivity. A better understanding of why deprived areas had more infections is required to more effectively reduce potential health and economic inequalities caused by higher infection rates in these areas compared to more affluent areas.

Deprivation and percentage of the area that was rural were not included in the main PCR and positivity models to avoid artificial

**Table 1 | Ranking of CIS areas in terms of swab positivity and the respective levels of deprivation in those areas**

| CIS area[a] | Region | Number of times with a >=80% probability of being in top 10 areas with high swab positivity (N = 74 weeks) | Average deprivation score rank of areas within the CIS area(1 most deprived)[b] | % of total population in the CIS area that lives in most deprived (10%) areas nationally[c] |
|---|---|---|---|---|
| Kirklees | Yorkshire | 23 | 47 | 12% |
| Nottingham | East Midlands | 16 | 5 | 31% |
| Rochdale | East Midlands | 20 | 8 | 30% |
| Burnley, Hyndburn, Pendle, Rossendale, Bury | North West | 15 | 24 | 22% |
| Manchester | North West | 10 | 2 | 43% |
| Newham | London | 9 | 22 | 2% |
| Barking and Dagenham | London | 8 | 11 | 4% |
| Blackburn with Darwen, Chorley, Bolton | North West | 8 | 23 | 23% |
| Bradford | Yorkshire | 8 | 7 | 34% |
| Salford | North West | 8 | 9 | 30% |

[a] Only the top 10 areas based on the frequency of the number of times with > 80% probability of being among the top 10 areas with high swab positivity are presented.

[b] Deprivation was measured using the 2019 English index of multiple deprivation at a smaller geographical level than CIS areas. The average deprivation score rank ranges from 1 (most deprived CIS area) to 116 (least deprived CIS area), estimated by ranking CIS areas based on the population-weighted average deprivation score of people living in different smaller areas – with each their deprivation score - within CIS areas.

[c] This percentage is based on first identifying the 10% most deprived areas in England as a whole based on the 2019 English index of multiple deprivation, and subsequently estimating the percentage of the total population in each CIS area that lives in those 10% most deprived areas in England.

associations from using the same variables to predict the prevalence in the target population and subsequently in a separate model to assess the relationship between the same variable and post-stratified prevalence. Furthermore, both variables are only readily available as area-based markers and partly already captured by post-stratifying to relatively small CIS areas within England. Ideally one would be able to account for individual-based socio-economic status related variables, such as educational qualifications and household assets, among survey participants and the general population (the target population) through linkage to the 2021 Census in England[13]. However, this was not available for research at the time of the current analysis. That a particular area ranks highly throughout a large part of the pandemic does not necessarily mean that the difference with other areas is large. However, the areas that most often ranked in the top 10 had 2 to 2.5 times higher PCR positivity than the average across all CIS areas during the weeks these areas had a high probability of being among the areas with the highest PCR positivity.

The validity of the post-stratification relies on the absence of model misspecification, e.g., not missing an important variable that both influences the decision to participate in the survey upon invitation and the outcomes considered here. Variables that are associated with the decision not to participate in surveys like this despite monetary compensation, which may increase the probability that those of lower socio-economic status participate[14], may also be associated with behaviour that increases the risk of acquiring an infection with SARS-CoV-2. This may have led to underestimating swab positivity, and hence antibody levels mediated through swab positivity, while characteristics that mainly affect outcomes through vaccination uptake are less relevant as we took into account vaccination status. Model complexity meant that we could only allow for whether participants were vaccinated as a binary variable (yes/no) interacting with time. More complicated models with the number of vaccinations failed to converge without errors due to very high/low positivity rates during large parts of the study period. Therefore, we implicitly assume that the survey - after conditioning on CIS area, age, sex, ethnicity, and time – may not be representative in terms of whether individuals decide to get any COVID-19 vaccine, but that - after conditioning on the same variables - vaccinated individuals in the survey are representative of vaccinated individuals in the target population.

Whether the post-stratification used effectively removes any bias due to non-response is also dependent on how accurate the information on conditional distributions of variables is in the target population. While the number of vaccinated individuals is well-recorded through the National Immunisation Management Service (NIMS) system, there is more uncertainty about the number of individuals in each subgroup of the population that did not get vaccinated as it is not exactly known how many people live in England. For the current study, we restricted to individuals that could be linked to the 2011 Census to ensure a well-defined denominator. Consequently, we had to make assumptions about the uptake of individuals aged 2–8 years old. However, the PCR positivity estimates for the 2–11 years old were not sensitive to the assumptions made, and we only evaluated antibody levels for those aged 16 years and above. The estimated number of unvaccinated individuals in each subgroup relies on the assumption that more recent migrants have similar rates of uptake as individuals of the same age, sex, ethnicity, and location as those that were already present in England during the 2011 Census. If this assumption does not hold, the administrative data on vaccination uptake may also be not completely accurate. As information from the 2021 Census is released, these may be updated.

While the use of the COVID-19 Infection Survey, which randomly invites individuals and test study participants at a fixed schedule independent of symptoms, is a clear strength, it inevitably leads to lower precision than using biased national testing data. To be able to obtain estimates at even smaller geographical levels than the CIS regions used here, one could attempt to remove the bias in the

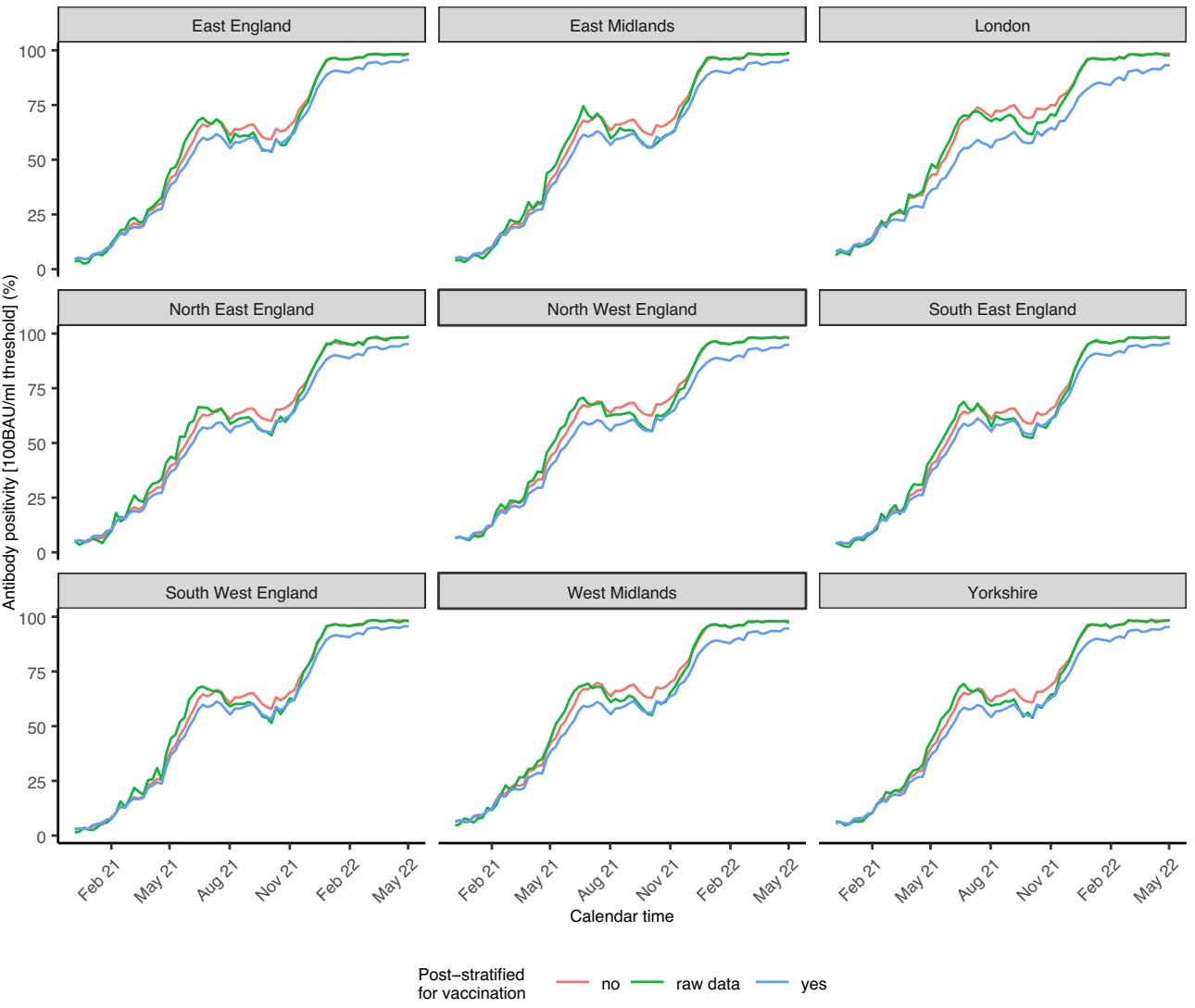

**Fig. 4 | Impact of post-stratifying for vaccination status (yes/no and interaction with time) on modelled estimated antibody positivity at the 100 BAU/ml threshold by region over time.** Modelled estimates are post-stratified for age, sex, CIS area, ethnicity and vaccination status. Estimates are presented as posterior medians (solid lines) with shading representing 95% credible intervals. Crude numbers from the underlying data are labelled as raw data. BAU: binding antibody units.

national testing data using the estimates provided here[4]. To obtain our estimates of PCR and antibody positivity over time and space, we extended an existing spatial regression and post-stratification approach by adding a time and space-time component[5]. This new approach has the benefits of being more efficient than fitting separate models to each week, it enabled us to account for temporal correlation in the panel data and to better quantify variation and uncertainty in (local) trends.

In conclusion, we have shown that not accounting for vaccination status overestimates the percentage of people that still have sufficient antibody levels to be protected against new infection, potentially affecting decision-making. Future analysis of the CIS and other surveys should, whenever possible, account for major drivers of the outcome of interest that are also likely associated with non-response to invitations to participate, with vaccination being particularly important when looking at infectious disease outcomes such as SARS-CoV-2 antibody levels. Using a spatiotemporal model that accounts for vaccination status, we have shown substantial variation between and within regions of England over the course of the pandemic and identified areas that had a high probability of having a higher SARS-CoV-2 prevalence than other areas throughout a large part of the pandemic. A

large part of the variation in the ranking of small areas in terms of their SARS-CoV-2 prevalence could be explained by the degree of urbanicity and deprivation, highlighting the inequality in risk of SARS-CoV-2 infections and its subsequent consequences.

## Methods

### Study participants

Data were obtained on all SARS-CoV-2 RT-PCR test results between 8 December 2020 and 04 May 2022 from nose and throat swabs taken from individuals participating in the Office for National Statistics (ONS) CIS (ISRCTN21086382, https://www.ndm.ox.ac.uk/covid-19/covid-19-infection-survey/protocol-and-information-sheets) living in private households in England. We restricted the current analyses to England, as detailed administrative data on vaccination uptake from which to construct post-stratification tables (see below) were only available for England. The survey randomly selects private households on an ongoing basis from address lists and previous surveys to provide a representative UK sample. Details on the sampling design are provided elsewhere[2]. Following verbal agreement to participate, a study worker visited each household to take written informed consent. This consent was obtained from parents/carers for those 2–15 years, while

those 10–15 years also provided written assent. Children aged < 2 years were not eligible for inclusion into the study.

Individuals were also asked about demographics and vaccination uptake (https://www.ndm.ox.ac.uk/covid-19/covid-19-infection-survey/case-record-forms). At the first visit, participants were asked for (optional) consent for follow-up visits every week for the next month, then monthly for 12 months from enrolment. Initially, in a random 10–20% of households, those 16 years or older were invited to provide blood monthly for assays of anti-trimeric spike protein IgG using an immunoassay developed by the University of Oxford[15,16]. Household members of participants who tested positive were also invited to provide blood monthly for follow-up visits. These participants were excluded from the analysis to avoid overestimation of antibody levels. From April 2021, additional participants were invited to provide blood samples monthly to assess vaccine responses, based on a combination of random selection and prioritisation of those in the study for the longest period (independent of test results)[10,17,18].

The study received ethical approval from the South Central Berkshire B Research Ethics Committee (20/SC/0195).

## Geographical units

England consists of 9 regions, formerly known as the government office regions. While they had partly devolved functions in the past, this is no longer the case. To be able to inform policy at a smaller geographic level than the national level, the ONS and government representatives divided England into 116 sub-regional areas that were deemed potentially relevant for local policymaking. These areas are nested within the 9 regions and consist of a single local authority district (LAD) in case the population of the LAD consists of at least 200,000 individuals, otherwise neighbouring LADs were combined into one subregional area. LADs, also known as local government districts, are used for the purposes of local government.

## Time units

The majority of survey participants were on a monthly visit schedule during the study period, given that the majority of individuals were recruited into the survey before the start of the study period. Given the large size of the survey, a large number of visits occur each day. In total 6,596,052 PCR tests were included during the study period of 74 weeks, meaning that on average there were on average 12,734 samples taken per day in England. However, given that we included 116 small areas and PCR positivity was at times well below 1%, we aggregated survey data into weeks to avoid model convergence issues encountered when modelling using daily data.

## Swab positivity

Nose and throat self-swabs were couriered directly to the UK's national Lighthouse laboratories (National Biocentre in Milton Keynes and Glasgow) where samples were tested as part of the national testing programme. The identical methodology was used to test for the presence of SARS-CoV-2 genes for nucleocapsid protein (N), spike protein (S), and ORF1ab using RT-PCR[2]. We used the TaqPath RT-PCR COVID-19 kit (Thermo Fisher Scientific, Waltham, MA, USA), which was analysed using UgenTec Fast Finder 3.300.5 (TagMan 2019-nCoV assay kit V2 UK NHS ABI 7500 v2.1; UgenTec, Hasselt, Belgium). The assay plugin contains an assay-specific algorithm and decision mechanism that allows the conversion of the qualitative amplification assay PCR raw data from the ABI 7500 Fast into test results with little manual intervention. Samples are called positive in the presence of at least one gene (N, ORF1ab, or both) but could be accompanied by the gene for S protein (ie, one, two, or three gene positives). The gene for S protein is not considered a reliable single gene positive[2]. For the purpose of comparison periods with different variants being dominant, the start dates of those periods were chosen as the first surveillance week (starting Monday) where > 50% of positive tests matched the S-gene of

the new variant (S-negative for Alpha (7 December 2020), Omicron BA.1 (13/12/2021); S-positive for Delta (17/05/2021) and Omicron BA.2 (21/02/2022).

## Antibody prevalence

Blood samples were couriered to the clinical biochemistry and microbiology laboratories at the John Radcliffe Hospital in Oxford to test for the presence of antibodies using the Oxford immunoassay[2,16]. Normalised results are reported in ng ml$^{-1}$ of mAb45 monoclonal antibody equivalents. Before 26 February 2021, the assay used fluorescence detection as described previously, with a positivity threshold of 8 million units validated on banks of known SARS-CoV-2 positive and SARS-CoV-2 negative samples[10,16].

After this, it used a commercialised CE-marked version of the assay, the OmniPATH 384 Combi SARS-CoV-2 IgG ELISA (Thermo Fisher Scientific), with the same antigen and colourimetric detection. mAb45 is the manufacturer-provided monoclonal antibody calibrant for this quantitative assay. The fluorometrically determined values were converted into arbitrary units using the following conversion formula:

$$\log_{10} = 0.221738 + 1.751889e - 07 * \text{fluorescence}_{\text{units}} + 5.416675e - 07 \\ * (\text{fluorescence}_{\text{units}} > 9190310) * (\text{fluorescence}_{\text{units}} - 9190310) \quad (1)$$

The results of the OmniPATH assay were converted into WHO international units (BAU ml$^{-1}$) using the following formula:

$$\text{BAU/mL} = 0.559 * [\text{mAb45 concentration in ng/mL at 1:50 dilution}] \quad (2)$$

The upper limit of quantification of the assay is 447 BAU/ml at the standard 1:50 dilution[10]. From 28 January 2022, samples were tested at 1:400 dilution, and from 29 April 2022 at 1:1600 dilution, with those below the lower limit of quantification at these dilutions retested at the original 1:50 dilution.

## Vaccination uptake

Participants were asked about their vaccination status at study visits, including information about the type of vaccine, the number of doses received to date, and the date of the most recent vaccination. For England, linked administrative vaccine uptake data is also available from the National Immunisation Management Service (NIMS), which records details of all COVID-19 vaccinations provided by the National Health Service (NHS) in England. NIMS covers the entire population of England but does not include vaccinations obtained abroad.

## Distribution of characteristics in the target population

The Office for National Statistics provided weekly estimates of the proportion of individuals in each subgroup of the population based on NIMS data linked to Census 2011 data. Therefore, only participants that were already living in England during the Census 2011 were used to inform conditional estimates of vaccine uptake, thereby implicitly assuming that vaccination uptake in each subgroup of the population is the same for individuals from that subgroup living in England as those from the same subgroup of the population that moved to England more recently. For children too young to be young to be present in the Census 2011 we assumed that no vaccination under the age of 5 and half the coverage of that observed among 9–11 years old children – who could be present in Census 2011 – for those aged 5–8 years of age. Different assumptions, such as assuming no vaccinations under the age of 9 did not materially affect vaccine uptake estimates given the fact that vaccination uptake among children aged 9–11 years of age, the ages from which we extrapolated to younger ages were very rarely vaccinated during the study period.

However, by linking the NIMS data to Census 2011 data at an individual level we could estimate vaccination uptake over time by all

variables considered here, including age, sex, ethnicity, and area. Estimates are available daily but aggregated by week in line with the grouping performed for the survey data. To be able to estimate the number of people in each so-called post-stratification cell, including those that were not present in Census 2011, e.g., the number of vaccinated 35–49 year old males of black ethnicity living in a private household in Northumberland in the 20[th] week of the study period, we applied these conditional estimates of vaccination uptake percentage to the estimated number of individuals living in private households in 2020 for the corresponding categories (age, sex, ethnicity, area). The latter estimates focused on individuals living in private households to line up with the sampling base of the CIS. These estimates were generated by ONS by updating the 2011 census using the cohort component method, ageing the population by 1 year each year, and incorporating births, deaths, immigration, emigration and people entering and leaving 'special populations' such as individuals in prisons[19].

The conditional distribution of ethnicity by these categories was obtained from the ETHPOP database[6,20]. Given that the ONS estimates do not come with a measure of uncertainty, uncertainty in the final post-stratifications table were not taken into account.

## Statistical analyses

Bayesian multilevel regression and poststratification is an increasingly used statistical technique to obtain representative estimates of prevalence or preferences at the national and smaller regional levels[2,5,21–29]. This method has been found superior at both national and regional levels compared to traditional survey-weighted and unweighted approaches in several empirical and simulation studies[2,5,21–29]. By using random effects in the multilevel model, stable estimates can be obtained for sub-national areas from relatively small samples or relatively rare outcomes[2]. However, if there is a spatial underlying structure this needs to be captured by the multilevel regression and post-stratification (MPR) methodology to avoid biased estimates based on a model that assumes independent group-level errors. Gao et al. recently proposed a spatial MRP model using a Besag-York-Mollié (BYM2) specification for the regional effect[5,30,31]. Using a simulation, they showed that when a spatial structure does exist, a spatial MRP with a BYM2 spatial prior for region improved MRP estimates through a reduction in absolute bias compared to using a default independent and identically distributed (IID) prior for the region effect. Importantly, when a spatial structure was not present in the underlying data, using a spatial MRP with a BYM2 spatial prior on region resulted in virtually identical posterior estimates as an MRP with an IID prior for region, suggesting that the BYM2 spatial prior does not force a spatial structure when it is not present[5].

Here we extend the spatial MRP approach proposed by Gao et al. to a spatio-temporal context by adding a temporal component to the model[5,6]. For the temporal components, we used random walk processes with discrete time indices (weeks) to capture likely temporal effects in the MRP model. The choice of the type of directed conditional distribution for the time effect (random walk or autoregressive) type of space-time interaction (type I–IV) and inclusion of additional covariates is guided by comparing the Watanabe-Akaike information criterion (WAIC) of the models[32,33]. A type I space-time interaction assumes no spatial and/or temporal structure on the interaction, a type II space-time interaction assumes that for the $i$th area the parameter vector has an autoregressive structure on the time component, which is independent of the ones of the other areas; a type III space-time interaction assumes that the parameters of the $t$th time point have a spatial structure independent from the other time points; while a type IV space-time interaction assumes that the temporal dependency structure for each area also depends on the temporal pattern of the neighbouring areas[32]. Sum-to-zero constraints appropriate for the type of interaction and the type of random walk (first- or second-order) used to model time were imposed to ensure the identifiability of the model in line with Goicoa

et al. and the bigDM R package (see also https://emi-sstcdapp.unavarra.es/bigDM/bigDM-3-fitting-spatio-temporal-models.html)[34,35]. Goicoa et al. explain in detail which identifiability constraints are necessarily dependent on how time is modelled and the type of interaction between space and time[34]. The appendix of that paper lists constraints required for different combinations of space-time interactions (I–IV) combined with first-order or second-order random walks for time[34]. Given that the best fitting model was a first-order random walk for time and a type-IV interaction between CIS area and time, below identifiability constraints were imposed[34].

$$\sum_{t=1}^{T} \delta_{it} = 0, \text{ for } i = 1, \ldots, S \quad (3)$$

$$\sum_{i=0}^{S} \xi_i = 0 \quad (4)$$

$$\sum_{t=1}^{T} \gamma_t = 0 \quad (5)$$

$$\sum_{i=1}^{S} \delta_{it} = 0, \text{ for } t = 1, \ldots, T \quad (6)$$

where $\xi$ represents the spatial random effect, $\gamma$ the temporal random effect (first-order random walk), and $\delta$ represents the structured interaction random effect (type IV interaction).

The same set of covariates and interactions were considered for the MRP models for all outcomes (swab positivity and antibody prevalence at different thresholds (23, 100, and 477 BAU/ml)) and included: age (2–11, 12–15, 16–24, 25–34, 35–49, 50–59, 60–64, 65–69, 70–74, 75–79, 80 +) (models for antibodies only included individuals aged 16 years or older given that blood samples were not taken in those aged < 16 y before November 2021); sex; ethnicity (Asian, Black, Mixed, White, Other); region (9 regions in England); CIS area (116 CIS areas, nested within regions in England); and two-way interactions of age, sex, and ethnicity with time and CIS or region area.

The best fitting model was a model with a first-order random walk for time (in weeks) and a type-IV interaction between CIS area and time, i.e., a model where the temporal dependency structure for each area also depends on the temporal pattern of the neighbouring areas. The linear predictor of this spatiotemporal model accounts additively for the temporal effects, spatial effects, and spatiotemporal interactions, and is linked to the expected value of the response $y$ through a logit link function, such that $E(y) = \text{logit}^{-1}(\eta)$.

For the final model with a first-order random walk for time and a type IV interaction between time and CIS area, we used penalised complexity (PC) priors - PC-prior(1,0.1) – for the precision of the random effects of covariates with multiple categories (ethnicity and age), the first-order random walk for time, and the space-time interaction. The precision of the ethnicity*time, age*time, and vaccination*time interactions was modelled using a PC-prior(1,0.01). For the other fixed effects (vaccinated [yes/no] and sex) we used priors with a mean of zero and precision of 0.01 (standard deviation of 10). Based on Gao et al – who found that the default BYM2 hyperprior specification for the mixing parameter $\varphi$ in INLA performed well in situations with and without a true spatial structure[5], we used the default PC-prior(0.5,0.5) for the mixing parameter of the BYM2 spatial prior.

For each outcome, after running the final spatiotemporal binomial regression model, post-stratification was used to obtain representative estimates of the outcome prevalence in the target population.

Using the population sizes of each post-stratification cell of the target population, MRP adjusts for residual non-representative by

post-stratifying by the percentage of each type in the actual overall population. If the model is correctly specified, unbiased estimates of prevalences at both national as well as sub-national and within categories can be obtained.

### Ranking of CIS areas in terms of prevalence

National testing data suggested that certain sub-regional areas in England had rather consistently higher prevalence of SARS-CoV-2 infections than the rest of the country, but it is not clear to what extent this is explained by testing behaviour or a true difference. Leveraging the fact that the applied models are Bayesian and the CIS is taking a random sample of the population, we evaluated the weekly probability that each CIS area is among the top 10 areas with the highest swab positivity prevalence.

Furthermore, we assessed to what extent the median ranking of CIS areas can be explained by area-specific levels of deprivation and degree of urbanity[36,37]. Area-level deprivation was based on the 2019 English index of multiple deprivation[17,38]. The average deprivation score rank ranges from 1 (most deprived CIS area) to 116 (least deprived CIS area), estimated by ranking CIS areas based on the population-weighted average deprivation score of people living in different smaller areas – with each their own deprivation score - within CIS areas. Linear regression with deprivation and urban/rural classification (modelled as a continuous variable) as covariates were used to evaluate to what extent these variables explain the median ranking of CIS areas in terms of swab positivity.

All analyses were performed in R version 4.1 using the following packages: ggplot (version 3.5.0); dplyr (version 1.1.4); tidyr; sp (version 2.1-3); rgdal (version 1.6-7); sf (version 1.0-16); INLA (version 23.09.09); arm (version 1.14-4); spdep (version 1.3-3); httr (version 1.4.7); tmap (version 3.3-4).

### Reporting summary

Further information on research design is available in the Nature Portfolio Reporting Summary linked to this article.

## Data availability

De-identified study data are available for access by accredited researchers in the ONS Secure Research Service (SRS) for accredited research purposes under part 5, chapter 5 of the Digital Economy Act 2017. Individuals can apply to be an accredited researcher using the short form on https://researchaccreditationservice.ons.gov.uk/ons/ONS_registration.ofml. Accreditation requires the completion of a short free course on accessing the SRS. To request access to data in the SRS, researchers must submit a research project application for accreditation in the Research Accreditation Service (RAS). Research project applications are considered by the project team and the Research Accreditation Panel (RAP) established by the UK Statistics Authority at regular meetings. Project application example guidance and an exemplar of a research project application are available. A complete record of accredited researchers and their projects is published on the UK Statistics Authority website to ensure transparency of access to research data. For further information about accreditation, contact Research.Support@ons.gov.uk or visit the SRS website.

## Code availability

A copy of the analysis code is available at https://github.com/pouwelskb/spatiotemporal_mrp_covid and https://doi.org/10.5281/zenodo.11109228[39].

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

## Acknowledgements

We are grateful for the support of all COVID-19 Infection Survey participants. This study is funded by the Department of Health and Social Care with in-kind support from the Welsh Government, the Department of Health on behalf of the Northern Ireland Government and the Scottish Government. A.S.W., T.E.A.P., N.S., D.W.E. and K.B.P. are supported by the National Institute for Health Research Health Protection Research Unit (NIHR HPRU) in Healthcare Associated Infections and Antimicrobial Resistance at the University of Oxford in partnership with the UK Health Security Agency (UKHSA) (NIHR200915). A.S.W. and T.E.A.P. are also supported by the NIHR Oxford Biomedical Research Centre. K.B.P. is also supported by the Huo Family Foundation. A.S.W. is also supported by core support from the Medical Research Council UK to the MRC Clinical Trials Unit [MC_UU_12023/22] and is an NIHR Senior Investigator. P.C.M. is funded by Wellcome (intermediate fellowship, grant ref 110110/Z/15/Z) and holds an NIHR Oxford BRC Senior Fellowship award. D.W.E. is supported by a Robertson Fellowship and an NIHR Oxford BRC Senior Fellowship. N.S. is an Oxford Martin Fellow and holds an NIHR Oxford BRC Senior Fellowship. The views expressed are those of the authors and not necessarily those of the National Health Service, NIHR, Department of Health, or UKHSA. This work contains statistical data from ONS which is Crown Copyright. The use of the ONS statistical data in this work does not imply the endorsement of the ONS in relation to the interpretation or analysis of the statistical data. This work uses research datasets which may not exactly reproduce National Statistics aggregates.

## Author contributions

The study was designed and planned by A.S.W., J.F., J.B., J.N., I.D. and K.B.P. and is being conducted by A.S.W., R.S., D.C., N.T., J.K., B.M., T.E.A.P., P.C.M., N.S., S.H., E.Y.J., D.I.S., D.W.C., D.W.E., T.H., and the COVID-19 Infection Survey Team. This specific analysis was designed by K.B.P. K. B. P. contributed to the statistical analysis of the survey data. B.A. contributed to the analysis of the administrative data. K.B.P. draughted the manuscript, and all authors contributed to the interpretation of the data and results and revised the manuscript. All authors approved the final version of the manuscript.

## Competing interests

D.W.E. declares lecture fees from Gilead, outside the submitted work. P.C.M. receives GSK funding to support a PhD fellowship in her team. The remaining authors declare no competing interests.

## Additional information

[1]Health Economics Research Centre, Nuffield Department of Population Health, University of Oxford, Oxford, UK. [2]The National Institute for Health Research Health Protection Research Unit in Healthcare Associated Infections and Antimicrobial Resistance at the University of Oxford, Oxford, UK. [3]Big Data Institute, Nuffield Department of Population Health, University of Oxford, Oxford, UK. [4]Department of Infectious Diseases and Microbiology, Oxford University Hospitals NHS Foundation Trust, John Radcliffe Hospital, Oxford, UK. [5]The National Institute for Health Research Oxford Biomedical Research Centre, University of Oxford, Oxford, UK. [6]Department of Mathematics, University of Manchester, Manchester, UK. [7]IBM Research, Hartree Centre, Sci-Tech, Daresbury, UK. [8]Office for National Statistics, Newport, UK. [9]The Francis Crick Institute, London, UK. [10]Nuffield Department of Medicine, University of Oxford, Oxford, UK. [11]Division of infection and immunity, University College London, London, UK. [12]European Centre for Environment and Human Health, University of Exeter, Truro, UK. [13]Office of the Regius Professor of Medicine, University of Oxford, Oxford, UK. [14]Wellcome Trust, London, UK. [15]Nuffield Department of Orthopaedics, Rheumatology and Musculoskeletal Sciences, University of Oxford, Oxford, UK. [16]MRC Clinical Trials Unit at UCL, UCL, London, UK.
✉e-mail: koen.pouwels@ndph.ox.ac.uk

## the COVID−19 Infection Survey Team

Jia Wei[3,10], Emma Pritchard[10], Karina-Doris Vihta[10], George Doherty[10], James Kavanagh[10], Kevin K. Chau[10], Stephanie B. Hatch[10], Daniel Ebner[10], Lucas Martins Ferreira[10], Thomas Christott[10], Wanwisa Dejnirattisai[10], Juthathip Mongkolsapaya[10], Sarah Cameron[10], Phoebe Tamblin-Hopper[10], Magda Wolna[10], Rachael Brown[10], Richard Cornall[10], Gavin Screaton[10], Katrina Lythgoe[3], David Bonsall[3], Tanya Golubchik[3], Helen Fryer[3], Tina Thomas[8], Daniel Ayoubkhani[8], Russell Black[8], Antonio Felton[8], Megan Crees[8], Joel Jones[8], Lina Lloyd[8], Esther Sutherland[8], Stuart Cox[17], Kevin Paddon[17], Tim James[17], Julie V. Robotham[18], Paul Birrell[18], Helena Jordan[19], Tim Sheppard[19], Graham Athey[19], Dan Moody[19], Leigh Curry[19], Pamela Brereton[19], Ian Jarvis[20], Anna Godsmark[20], George Morris[20], Bobby Mallick[20], Phil Eeles[20], Jodie Hay[21], Harper VanSteenhouse[21], Jessica Lee[22], Sean White[23], Tim Evans[23], Lisa Bloemberg[23], Katie Allison[24], Anouska Pandya[24], Sophie Davis[24], David I. Conway[25], Margaret MacLeod[25] & Chris Cunningham[25]

[17]Oxford University Hospitals NHS Foundation Trust, Oxford, UK. [18]UK Health Security Agency, London, UK. [19]IQVIA, London, UK. [20]National Biocentre, Milton Keynes, UK. [21]Glasgow Lighthouse Laboratory, London, UK. [22]Department of Health and Social Care, London, UK. [23]Welsh Government, Cardiff, UK. [24]Scottish Government, Edinburgh, UK. [25]Public Health Scotland, Edinburgh, UK.

