## [Peer Review File · Nature Communications]

Improving the representativeness of UK's national COVID-19 Infection Survey through spatio-temporal regression and post-stratificationREVIEWER COMMENTS

Reviewer #1 (Remarks to the Author):

The strength of this manuscript is the careful implementation of spatial and temporal modeling of survey data (even though the text dedicated to this point is quite minor). While there has been a growing body of studies using MRP and post-stratification weighting for surveys like the CIS, there are very few studies which have included both spatial correlations and temporal modeling together for small areas. As long as the authors can provide the code and sample data with publication, this contribution of the paper seems valuable.

However, I cannot support the publication of this current manuscript for other reasons. I list my comments below, but the most fundamental one is that I cannot really see a unified point of the various results and exercises from the estimates. The writing feels unfocused. What is the policy/medical/academic motivation for trying to obtain estimates of these outcomes by each of the 116 areas? How would this change existing public health / methods practice? The current intro only asserts the usefulness of having a survey rather than admin data, and does not explain the motivation for the analyses the authors conduct. Then, in the results the paper goes in and out of analyzing rankings within regions, correlations with depravity, and rurality --- I cannot catch the thread that links these variables. I am not a public health researcher so I may be missing something obvious to that community, but the audience for these findings seem small.

To summarize: The method itself seems sound, but if the contribution of this article were to be methodological, I think there should be more methodological discussion, comparison of different alternative approaches (such as the ones I list below), and documentation.

Other questions and suggestions follow.

Method Overview:

One important question – in the estimates “without vaccination adjustment”, do the authors also do a MRP with all the other variables? To believe that the claim that these differences shown should be attributed to weighting on vaccination, it must be the case that the estimates without vaccination should have the same model – MRP, weighting on demographics and area --- except for vaccination. This needs to be made explicit.

Two methodological requests: (1) How large are the standard errors (i.e. the standard deviations of the posterior estimates) of each area estimate? There is no barely any discussion of the uncertainty in these estimates, which is a drawback for survey data compared to administrative data. (2) How do the MRP estimates of regions differ from raw estimates without any weighting, and weighted estimates without MRP (for the main results)? The reason I ask is that the sample sizes here are quite large and so small area estimation may not be necessary.

On the framing: I'm not sure with the methodological contribution of the paper. The abstract and results talk about post-stratifying to vaccination rates as the main contribution (as in Bradley et al.). However, there are only two figures about this, and their methods describe the authors running a MRP with time-space varying components and various tests for correlated error. Of course, poststratifying is a part of MRP, but I feel this MRP method that the authors implement is equally if not a more important methodological addition than simply weighting to vaccination.

Method Implementation

- More detail is needed on what units the poststratification was done for. Does the survey come in waves (e.g. a large chunk in one week) or do the samples come in daily? If the latter, how well does the vaccination target match up to the survey date, per day? Are the denominators of the administrative vaccination numbers and the survey sampling frame the sample? See the Bradley et al. article in *Nature*, especially its methodological section, for how this was dealt with in the US context.

- Is vaccination available for each of the 116 regions, or is it available for other subgroups? The paper says the distributions are "conditional" but it does not say conditional on what. If the full joint table is not available, how do the authors estimate a full poststratification table (e.g. the number of adults that are in London, male, Black, _and_ age 18-24)? Do the authors think these demographics are independent from each other? That would be reasonable given data limitations, but it should be defended.

Specific comments on figures and tables follow. Overall, the ink spent on many of the figures and tables do not seem to be proportional to the findings that are in the text.

Fig. 1:

- The colored lines are hard to tell apart. Consider facets instead?
- The right panel appears unnecessary, as it is never mentioned in the text and I cannot understand what it is supposed to show or what the point is. No information on units either.

Fig. 2:

- "6.30% vs 6.69%" is 0.38 percentage *points*, not 0.38 percent as the manuscript says. This should be fixed throughout the manuscript. The distinction can become consequential in determining what is large or small. For example, the difference between 0.10% and 0.20% is 0.10 percentage points but also 100 percent. Is this a small or large discrepancy? The author should justify either with public health context.

- The discussion about children in the caption/legend of Fig.2 cannot be understood without reading way into the discussion (and even then the description is unclear) – this text does not belong in Fig.2.
- The default of ggplot should be changed so that outcomes that cannot take a negative value do not show negative space below the zero in the plot.

Fig. 3

- Changing the axes by wave makes the cross-date interpretation impossible. I think the authors should use the same legend throughout.

Fig. 4 and 7

- These plots are too messy and I cannot see what the takeaway from this plot might be. More specifically: Showing the outcome in terms of rank (within regions) is not substantively meaningful. It can also exaggerate the degree of variation because a small difference between areas can lead to a huge difference in ranking. By ranking within region, there is no cross-region interpretation possible either. There should also be more information on how the IQR is computed.

Table 1

- I do not understand the use of "deprivation" in this table and throughout the text. What is the motivation, how is it measured, etc? There should be a caption or legend for Table 1.

Reviewer #2 (Remarks to the Author):

The topic is important, especially looking forward to future epidemics. The statistical approach seems reasonable, and the article makes a convincing case for why it was necessary.

Reviewer #3 (Remarks to the Author):

This manuscript applies multilevel regression and poststratification (MRP) to adjust demographics and vaccination status of respondents in the UK's national COVID-19 Infection Survey (CIS) and generate representative viral infection prevalence and antibody levels. The basic idea is not new, as Covello et al. (2021) and Si et al. (2021) have applied MRP to hospital testing data and obtained synthetic random proxy metrics. These two papers were not mentioned in the manuscript. As an extension, the authors include spatio-temporal structure under MRP. However, it is unclear whether such an extension offers benefits. In the Section of Statistical analyses, it says "The choice of the type of directed conditional distribution for the time effect (random walk or autoregressive) type of space-time interaction (type I-IV) and inclusion of additional covariates is guided by comparing the Watanabe-Akaike information criterion (WAIC) of the models." It is unclear what the final structure is and the extent to which it performs better than alternatives. How much smoothness or shared information the spatio-temporal structure has offered to stabilize the neighborhood estimates?

Since the UK's CIS depends on voluntary participation, the manuscript emphasizes that vaccination status is an important confounder to be adjusted for in terms of selection bias. What about COVID-related symptoms? Will individuals have symptoms have higher tendency to participate the CIS than those without symptoms? If so, symptom status should also be accounted for.

Area-specific covariates, such as levels of deprivation and rurality percentages, are used for exploratory analyses after model estimation. Are they predictive covariates in the model? In the Discussion, the manuscript says they are not included in the model to avoid duplicated use. However, since the goal is to obtain stabilized spatial estimates, spatial covariates would play an important role in the model to avoid over-smoothing.

In the Section of Statistical analysis, it writes "Post-stratification tables were based on the conditional distribution of age and sex by area from ONS. The conditional distribution of ethnicity by these categories were obtained from the ETHPOP database.^{5,27} Conditional distribution of vaccination uptake were obtained from the NIMS administrative data on vaccination uptake." This means that there is no existing joint distribution of age, sex, area, ethnicity, and vaccination status in the "population". How were these separate conditional distributions merged into a joint table? What are the assumptions? Will the uncertainty be accounted for?

References

Covello L, Gelman A, Si Y, Wang S. Routine Hospital-based SARS-CoV-2 Testing Outperforms State-based Data in Predicting Clinical Burden. *Epidemiology*. 2021 Nov 1;32(6):792-799. doi: 10.1097/EDE.0000000000001396. PMID: 34432721; PMCID: PMC8478110.

Si Y, Covello L, Wang S, Covello T, Gelman A. Beyond Vaccination Rates: A Synthetic Random Proxy Metric of Total SARS-CoV-2 Immunity Seroprevalence in the Community. *Epidemiology*. 2022 Jul 1;33(4):457-464. doi: 10.1097/EDE.0000000000001488. Epub 2022 Mar 29. PMID: 35394966; PMCID: PMC9148633.

Reviewer #4 (Remarks to the Author):

Summary

This manuscript adds to evidence that accurate, representative, small-area estimates of the prevalence of COVID-19 are unusual and hard to produce. The key issue is selective participation (or non-participation) that is related to the outcome(s) of interest - COVID-19 status of some sort. Prior work has revealed that selective non-response to erstwhile population-representative surveys can introduce important bias in estimated prevalence - e.g. political ideology associated with

behaviors that affect COVID-19 status (PNAS, <https://doi.org/10.1073/pnas.202394711>). This work uses the same approach to evaluate the effect of not accounting for vaccination status when estimating infection prevalence and antibody levels using high quality, longitudinal data. The results demonstrate that accounting for vaccination status is essential, but that the effect is nuanced as the epidemic progresses and vaccination campaigns are increasingly successful.

Data Methods

The data are unusually useful - high quality and sufficient quantity. With respect to methods, the results further validate an approach that involves model-based estimation using Bayesian hierarchical space-time models coupled with post-stratification to produce population-level estimates. Over the past few years this combination of methods has been shown to be generally superior in comparison to traditional frequentist estimation approaches, more robust, and the only viable approach when data are defective in various ways.

COVID-19 Status Estimates

With respect to producing large and small-area estimates of COVID-19 status, this work clearly demonstrates that including vaccination status in the models is essential; potentially large, consequential errors are possible when vaccination status is ignored.

Presentation and Reproducibility

The work is presented well in text and figures and there is a copy of the R script that was used for analysis. However, I could not find the statistical models or their priors written out anywhere in mathematical notation. This makes it hard or essentially impossible to completely interrogate or understand exactly what was done (or, it requires poring over the code very carefully, which I did not do). I strongly suggest that the statistical models and associated material (e.g. priors) be written out clearly using standard notation.

Challenges

As far as I can tell, the main challenge with this work is the 2011 baseline census to determine the size of each cell. I wonder if it would be possible to conduct a sensitivity analysis of some sort to gauge exactly how consequential this issue may be - a decade of population change could change things significantly.

Work Referenced

You may want to reference "Estimating seroprevalence of SARS-CoV-2 in Ohio: A Bayesian multilevel poststratification approach with multiple diagnostic tests" (<https://doi.org/10.1073/pnas.202394711>) that applied the same method to a similar situation with the added complexity of multiple tests but without the temporal dimension.

Response to the reviewers:

REVIEWER COMMENTS

Reviewer #1 (Remarks to the Author):

1. The strength of this manuscript is the careful implementation of spatial and temporal modeling of survey data (even though the text dedicated to this point is quite minor). While there has been a growing body of studies using MRP and post-stratification weighting for surveys like the CIS, there are very few studies which have included both spatial correlations and temporal modeling together for small areas. As long as the authors can provide the code and sample data with publication, this contribution of the paper seems valuable.

Response: Thank you for highlighting our important contribution to the literature. And as indicated in the manuscript we will make code available upon publication via github (and the code was available to the reviewers as we have added that when we submitted our manuscript originally).

2. However, I cannot support the publication of this current manuscript for other reasons. I list my comments below, but the most fundamental one is that I cannot really see a unified point of the various results and exercises from the estimates. The writing feels unfocused. What is the policy/medical/academic motivation for trying to obtain estimates of these outcomes by each of the 116 areas? How would this change existing public health / methods practice? The current intro only asserts the usefulness of having a survey rather than admin data, and does not explain the motivation for the analyses the authors conduct. Then, in the results the paper goes in and out of analyzing rankings within regions, correlations with depravity, and rurality --- I cannot catch the thread that links these variables. I am not a public health researcher so I may be missing something obvious to that community, but the audience for these findings seem small.

Response: We clarified in the introduction why the 116 areas were used:

"Early in the pandemic, the Office for National Statistics (ONS) and government representatives from England, Northern-Ireland, Scotland and Wales divided the UK in 133 sub-regional areas that were deemed relevant for local policy making and simultaneously sufficiently large to provide meaningful estimates of swab positivity and antibody prevalence. For the current analysis we focus on the 116 areas in England given the availability of detailed administrative data on vaccination uptake during the study period."

In addition, we added more details about the used geographies in the methods section:

"Geographies

England consists of 9 regions, formerly known as the government office regions. While they had partly devolved functions in the past, this no longer the case. To be able to inform policy at a smaller geographic level than the national level, the ONS and government representatives divided England into 116 sub-regional areas that were deemed potentially relevant for local policy making. These areas are nested within the 9

regions and consist of a single local authority district (LAD) in case the population of the LAD consists of at least 200,000 individuals, otherwise neighbouring LADs were combined into one subregional area. LADs, also known as local government districts, are used for the purposes of local government.”

To further clarify the rationale behind the incorporated analyses we added the following: “In addition, given the large interest of policy makers in understanding why certain areas in England consistently had higher number of people testing positive in the national testing programme, we evaluated whether this observation might be explained by regional variation in the probability of deciding to take a test upon symptoms, or whether similar trends were observed in the CIS where survey participants are tested based on a fixed schedule independent of symptoms status. Furthermore, given the fact that regional variation in a long list of health outcomes and behaviours can be explained by deprivation and urbanicity, we evaluated to what extent areas that frequently have higher swab positivity estimates compared to other areas are more deprived and more urban.”

However, we agree with the reviewer that there may be less interest in the ranking of antibody prevalence, reflected by policy makers regularly asking why certain areas consistently had a higher swab positivity, while rarely asking why antibody prevalence is different. For the latter there are 2 main drivers: vaccination uptake and infection rates, which are either known through the administrative data (vaccination data) or already reflected in the ranking of positivity (which is already covered by the analysis focusing on swab positivity). As such we removed the paragraphs on the ranking of the antibody prevalence, but kept the paragraphs focusing on ranking of swab positivity.

3. To summarize: The method itself seems sound, but if the contribution of this article were to be methodological, I think there should be more methodological discussion, comparison of different alternative approaches (such as the ones I list below), and documentation.

Response:

We have added more discussions and comparisons, as detailed in the responses to the comments below. In addition, to be able to focus more on the methods and incorporate other suggestions we have also removed the supplementary analyses at different antibody thresholds and only included results for the 100 BAU threshold.

Method Overview:

4. One important question – in the estimates “without vaccination adjustment”, do the authors also do a MRP with all the other variables? To believe that the claim that these differences shown should be attributed to weighting on vaccination, it must be the case that the estimates without vaccination should have the same model – MRP, weighting on demographics and area --- except for vaccination. This needs to be made explicit.

Response: We have made more clear now that the model with and without vaccination adjustment are indeed identical, except the in/exclusion of the vaccination variable (and it's interaction with time). This was already detailed in the methods and discussion section, but we have added this information more clearly in the results section as well.

We added the following in the results section:

"The best fitting spatiotemporal multilevel regression model for PCR positivity, based on the Watanabe-Akaike information criterion (WAIC), was a model that included terms for age, sex, ethnicity, vaccination status, a Besag-York-Mollié (BYM2) specification for the CIS area effect, and two-way interactions between time – measured in weeks – and age, ethnicity, vaccination status, and CIS area. The only covariate term that did not lead to clear improvement in the WAIC was the interaction between sex and time. Inclusion of the latter term resulted in an increase instead of the decrease of WAIC of ≥ 20 observed for other covariates. The interaction between CIS area and time was modelled using a so-called type IV space-time interaction, assuming that for the i th area the parameter vector has an autoregressive structure on the time component and that each time point there is spatial correlation. The combination of a type IV space-time interaction, whereby time was modelled using a first-order random walk was chosen based on the WAIC, having a better model fit than other types of interactions and/or other ways to model time (WAIC difference ≥ 30).

While survey participants were more likely to be vaccinated than expected based on the national administrative data on vaccination uptake, adding an indicator for vaccination status (yes/no) interacting with time in weeks to a model that already accounted for all other variables improved model fit (WAIC difference of 964), but this had only small effects on post-stratified estimated levels of PCR positivity (Fig. 2)."

In addition, we have added a line in figure 2 representing the raw, unadjusted PCR positivity over time. We also added the following:

"The unadjusted positivity among survey participants led to even larger underestimations of PCR positivity until the Omicron BA2 peak when there were shifts in terms of ethnic and age-related risk groups (Fig. 2, Fig. S1, Fig. S2)."

5. Two methodological requests: (1) How large are the standard errors (i.e. the standard deviations of the posterior estimates) of each area estimate? There is no barely any discussion of the uncertainty in these estimates, which is a drawback for survey data compared to administrative data. (2) How do the MRP estimates of regions differ from raw estimates without any weighting, and weighted estimates without MRP (for the main results)? The reason I ask is that the sample sizes here are quite large and so small area estimation may not be necessary.

Response: The 95% credible intervals, as a standard measure of uncertainty were all already plotted in the figures and detailed in Table S1. Because policy makers tend to find credible intervals more intuitive than standard deviations, in our experience communicating the survey results to UK policy makers throughout the pandemic using

the methodology described in this paper, we used the former to express the degree of uncertainty.

To further emphasise the degree of precisions we have now added a sentence on the average 95% credible interval width, which of course is also dependent on the prevalence itself and as such varies over time (as would the standard deviations):

"Post-stratified estimates of the fully adjusted model including 95% credible intervals (CrI) aggregated by England, region, CIS area, age, and ethnicity are provided in table S1. While CrI width varies over time and depends on the PCR positivity, at the smallest geographic CIS level the average 95% CrI width was 1.08 pp."

In addition, we added a comparison with crude data in the figures comparing the model MRP estimates with and without vaccination for swab and antibody positivity, and added the following sentence regarding swab positivity: "The underestimation of swab positivity was more pronounced for crude swab positivity based on the raw data during most of the study period." For antibody positivity we added the following: "Similarly, using crude positivity estimates generally resulted in an overestimation during periods that the MRP model that did not account for vaccination status led to overestimates of antibody positivity."

Furthermore, we clarified that "many areas had weeks with 2 or less positive samples in total despite the size of the survey (25th percentile of the number of positive tests per week was 2), emphasising the need for small-area estimation and potentially explaining why a better model fit was obtained for models with a type IV interaction and a random walk structure for time." Despite the size of the survey there are hence insufficient number of positives to obtain reliable weighted estimates without MRP/small area estimation.

In addition, we added the following to the discussion:

"While the use of the COVID-19 Infection Survey, which randomly invites individuals and test study participants at a fixed schedule independent of symptoms, is a clear strength, it inevitably leads to less precisions than using biased national testing data. To be able to obtain estimates at even smaller geographical levels than the CIS regions used here, one could attempt to remove the bias in the national testing data using the estimates provided here."

6. On the framing: I'm not sure with the methodological contribution of the paper. The abstract and results talk about post-stratifying to vaccination rates as the main contribution (as in Bradley et al.). However, there are only two figures about this, and their methods describe the authors running a MRP with time-space varying components and various tests for correlated error. Of course, poststratifying is a part of MRP, but I feel this MRP method that the authors implement is equally if not a more important methodological addition than simply weighting to vaccination.

Response: We would first note that Bradley et al did NOT post-stratify to vaccination rates, they tried to estimate vaccination uptake, but did not use that in an attempt to

estimate anything that is not already known through admin data, such as PCR positivity or antibodies.

We have added more detail on the impact of post-stratifying and the addition of time-space varying components:

Intro:

"Here, we use data from the UK's national COVID-19 Infection Survey (CIS) to demonstrate how a spatio-temporal regression and post-stratification modelling approach, extending previously developed spatial regression and post-stratification using a time and space-time component, can be used to obtain representative temporal estimates of the swab positivity and antibody prevalence at the national and sub-regional level, and for different ages and ethnicities."

Results:

"The best fitting spatiotemporal multilevel regression model for PCR positivity, based on the Watanabe-Akaike information criterion (WAIC), was a model that included terms for age, sex, ethnicity, vaccination status, a Besag-York-Mollié (BYM2) specification for the CIS area effect, and two-way interactions between time – measured in weeks – and age, ethnicity, vaccination status, and CIS area. The only covariate term that did not lead to clear improvement in the WAIC was the interaction between sex and time. Inclusion of the latter term resulted in an increase instead of the decrease of WAIC of ≥ 20 observed for other covariates. The interaction between CIS area and time was modelled using a so-called type IV space-time interaction, assuming that for the i th area the parameter vector has an autoregressive structure on the time component and that each time point there is spatial correlation. The combination of a type IV space-time interaction, whereby time was modelled using a first-order random walk was chosen based on the WAIC, having a better model fit than other types of interactions and/or other ways to model time (WAIC difference ≥ 30).

While survey participants were more likely to be vaccinated than expected based on the national administrative data on vaccination uptake, adding an indicator for vaccination status (yes/no) interacting with time in weeks to a model that already accounted for all other variables improved model fit (WAIC difference of 964), but this had only small effects on post-stratified estimated levels of PCR positivity (Fig. 2)."

Discussion:

"To obtain our estimates of PCR and antibody positivity over time and space, we extended an existing spatial regression and post-stratification approach by adding a time and space-time component. This new approach has the benefits of being more efficient than fitting separate models to each week, it enabled us to account for temporal correlation in the panel data and to better quantify variation and uncertainty in (local) trends."

Method Implementation

7. - *More detail is needed on what units the poststratification was done for. Does the survey come in waves (e.g. a large chunk in one week) or do the samples come in daily? If the latter, how well does the vaccination target match up to the survey date, per day? Are the denominators of the administrative vaccination numbers and the survey sampling frame the sample? See the Bradley et al. article in *Nature*, especially its methodological section, for how this was dealt with in the US context.*

Response:

We clarified the time units available and used for analyses:

"Time units

The majority of survey participants were on monthly visit schedule during the study period, given that the majority of individuals were recruited into the survey before the start of the study period. Given the large size of the survey, a large number of visits occur each day. In total 6,596,052 PCR tests were included during the study period of 74 weeks, meaning that on average there were on average 12,734 samples taken per day in England. However, given that we included 116 small areas and PCR positivity was at times well below 1%, we aggregated survey data into weeks to avoid model convergence issues encountered when modelling using daily data."

Furthermore, we signalled where the text about the distribution of characteristics in the target population.

"Distribution of characteristics in the target population

The Office for National Statistics provided weekly estimates of the proportion of individuals in each subgroup of the population based on NIMS data linked to Census 2011 data. Therefore, only participants that were already living in England during the Census 2011 were used to inform conditional estimates of vaccine uptake, thereby implicitly assuming that vaccination uptake in each subgroup of the population is the same for individuals from that subgroup living in England as those from the same subgroup of the population that moved to the England more recently. However, by linking the NIMS data to Census 2011 data at an individual level we can estimate vaccination uptake over time by all variables considered here, including age, sex, ethnicity, and area. Estimates are available on a daily basis, but aggregated by week in line with the grouping performed for the survey data. To be able to estimate the number of people in each so-called post-stratification cell, e.g. the number of vaccinated 35-49 year old males of black ethnicity living in a private household in Northumberland in the 20th week of the study period, we applied these conditional estimates of vaccination uptake percentage to the estimated number of individuals living in private households in 2020 for the corresponding categories. The latter estimates focused on individuals living in private households to line up with the sampling base of the CIS. These estimates were generated by ONS by updating the 2011 census using the cohort component method, aging the population by 1 year each year, and incorporating births, deaths, immigration, emigration and people entering and leaving 'special populations' such as individuals in prisons."

8. - Is vaccination available for each of the 116 regions, or is it available for other subgroups? The paper says the distributions are "conditional" but it does not say conditional on what. If the full joint table is not available, how do the authors estimate a full poststratification table (e.g. the number of adults that are in London, male, Black, _and_ age 18-24)? Do the authors think these demographics are independent from each other? That would be reasonable given data limitations, but it should be defended.

Response:

We have now further clarified that vaccination uptake is available by all other variables, see the response above ('distribution of characteristics in the target population' heading).

9. Specific comments on figures and tables follow. Overall, the ink spent on many of the figures and tables do not seem to be proportional to the findings that are in the text.

Response: We have now added more details regarding what is shown in the figures and tables. In addition, we removed some figures in a response to reviewer comments as well as merged others. Furthermore, we replaced table S2 with one sentence summarising the point we made about this table.

10. # Fig. 1:

- The colored lines are hard to tell apart. Consider facets instead?

- The right panel appears unnecessary, as it is never mentioned in the text and I cannot understand what it is supposed to show or what the point is. No information on units either.

Response: We removed the right hand panel of this figure and used the extra space created to indeed include a facet plot.

11. # Fig. 2:

- "6.30% vs 6.69%" is 0.38 percentage *points*, not 0.38 percent as the manuscript says. This should be fixed throughout the manuscript. The distinction can become consequential in determining what is large or small. For example, the difference between 0.10% and 0.20% is 0.10 percentage points but also 100 percent. Is this a small or large discrepancy? The author should justify either with public health context.

Response: We have changed % to percentage points where applicable. We already previously discussed in the discussion whether differences were large or small:

"Not accounting for vaccination status, as done throughout the pandemic, resulted in a small underestimation of PCR positivity but a more substantial overestimation of the percentage of the population having antibody levels at least as high as the threshold previously estimated to be associated with 67% protection against infection with the Delta variant. While estimates were, as expected, similar with and without accounting for vaccination status at the start of the vaccination campaign, and 5.2 pp% when taking the average difference for England over the entire period, the maximum difference in antibody prevalence at the aforementioned antibody threshold was 21 pp%. Such large differences, observed in an area with one of the lowest percentages of White British

populations and one of the highest poverty rates in England, could lead to underinvestment in interventions that could help increase the percentage of the local population with sufficiently high antibody levels to be protected against new SARS-CoV-2 infections, potentially contributing to severe acute disease or long COVID-19 in the population.”

12. - The discussion about children in the caption/legend of Fig.2 cannot be understood without reading way into the discussion (and even then the description is unclear) – this text does not belong in Fig.2.

- The default of ggplot should be changed so that outcomes that cannot take a negative value do not show negative space below the zero in the plot.

Response:

We have added the information from the legend to the methods section and changed it to make it more clear:

“For children too young to be present in the Census 2011 we assumed that no vaccination under the age of 5 and half the coverage of that observed among 9-11 years old children – who could be present in Census 2011 – for those aged 5-8 years of age. Different assumptions, such as assuming no vaccinations under the age of 9 did not materially affect vaccine uptake estimates given the fact that vaccination uptake among children aged 9-11 years of age, the ages from which we extrapolated to younger ages were very rarely vaccinated during the study period.”

In addition we changed the plots as suggested.

13. # Fig. 3

- Changing the axes by wave makes the cross-date interpretation impossible. I think the authors should use the same legend throughout.

Response: We do not agree that the axes by timepoint should be kept the same, the intention of this plot is to show the variation between areas at different time point, which is best illustrated using different legends given the markedly changing prevalence over time. Together with the line figures and supplementary table S1 – which can be redrawn and used by anyone who intends to further understand the underlying figures – this gives a complete picture of variation by space and by time.

14. # Fig. 4 and 7

- These plots are too messy and I cannot see what the takeaway from this plot might be. More specifically: Showing the outcome in terms of rank (within regions) is not substantively meaningful. It can also exaggerate the degree of variation because a small difference between areas can lead to a huge difference in ranking. By ranking within region, there is no cross-region interpretation possible either. There should also be more information on how the IQR is computed.

Response:

As explained in our response to the other reviewer, we included the ranking as this was a specific question from policy makers. However, we removed figure 7 (see response to reviewer 1).

As already indicated in the legend, the ranking is compared to other CIS areas in the country, not within a particular region. "A low ranking corresponds to a CIS area having lower swab positivity estimates at that point in time compared to other CIS areas in the country." Thus, this figure facilitates cross-interpretation.

However, given that the ranking of areas is also already captured by table 1, and the variation over time in prevalence by figure 3 and S3, we also removed figure 4 as we could not find a way to make the plots less 'messy' (the data are just messy in areas where areas do not rank very clearly high or low throughout the pandemic). The figure is much less 'messy' in the South-West region, where it is very obvious that the South-west regions were initially the areas with the lowest PCR positivity, whereas during Omicron dominant periods they were highest. Of note the IQR was based on the posterior draws (as is the 80% probability), evaluating the ranking within each draw and taking the 25th and 75th percentile.

Given that policy makers were particularly interested in the ranking of areas and were confused about how to interpret e.g. a time-varying z-score, we kept our analysis based on ranking (but removed figure 4 as suggested by the reviewer), but added the following to the result section to clarify that important differences occurred between areas that rank highly and those that don't:

"During the times that the Kirklees and Rochdale areas had a high probability of being ranked in the top 10 areas of highest swab positivity, their posterior mean estimates were on average, respectively, 2 and 2.5 times higher than the average of all 116 CIS areas during those weeks."

And to the discussion section:

"That a particular area ranks highly throughout a large part of the pandemic does not necessarily mean that the difference with other areas is large. However, the areas that most often ranked in the top 10 had 2 to 2.5 times higher PCR positivity than the average across all CIS areas during the weeks these areas had a high probability of being among the areas with the highest PCR positivity."

15. ## Table 1

- I do not understand the use of "deprivation" in this table and throughout the text. What is the motivation, how is it measured, etc? There should be a caption or legend for Table 1.

Response:

We have added more detail about why we focused on deprivation in the introduction and how deprivation was measured in the methods section (see also responses above).

In addition we added information regarding deprivation in the table caption and legend.

Reviewer #2 (Remarks to the Author):

1. The topic is important, especially looking forward to future epidemics. The statistical approach seems reasonable, and the article makes a convincing case for why it was necessary.

Response: We thank the reviewer for their positive review.

Reviewer #3 (Remarks to the Author):

1. This manuscript applies multilevel regression and poststratification (MRP) to adjust demographics and vaccination status of respondents in the UK's national COVID-19 Infection Survey (CIS) and generate representative viral infection prevalence and antibody levels. The basic idea is not new, as Covello et al. (2021) and Si et al. (2021) have applied MRP to hospital testing data and obtained synthetic random proxy metrics. These two papers were not mentioned in the manuscript. As an extension, the authors include spatio-temporal structure under MRP. However, it is unclear whether such an extension offers benefits. In the Section of Statistical analyses, it says "The choice of the type of directed conditional distribution for the time effect (random walk or autoregressive) type of space-time interaction (type I-IV) and inclusion of additional covariates is guided by comparing the Watanabe-Akaike information criterion (WAIC) of the models." It is unclear what the final structure is and the extent to which it performs better than alternatives. How much smoothness or shared information the spatio-temporal structure has offered to stabilize the neighborhood estimates?

Response:

We have added the references to the text, but do note that the methods developed by us (the time extension) and described in this paper were taken up by the Office for National Statistics – while not accounting for vaccination – for sharing results on sub-regional COVID-19 infections to policy makers on a weekly basis from November 2020 onwards, well before the referenced papers were published.

We have added more detail on the addition of the spatiotemporal structure as well as indications of the degree of improved model fit using the difference in WAIC:

"Here, we use data from the UK's national COVID-19 Infection Survey (CIS) to demonstrate how a spatio-temporal regression and post-stratification modelling approach, extending previously developed spatial regression and post-stratification using a time and space-time component, can be used to obtain representative temporal estimates of the swab positivity and antibody prevalence at the national and sub-regional level, and for different ages and ethnicities."

Results:

"The best fitting spatiotemporal multilevel regression model for PCR positivity, based on the Watanabe-Akaike information criterion (WAIC), was a model that included terms for age, sex, ethnicity, vaccination status, a Besag-York-Mollié (BYM2) specification for the CIS area effect, and two-way interactions between time – measured in weeks – and age, ethnicity, vaccination status, and CIS area. The only covariate term that did not lead to clear improvement in the WAIC was the interaction between sex and time. Inclusion of the latter term resulted in an increase instead of the decrease of WAIC of ≥ 20 observed for other covariates. The interaction between CIS area and time was modelled using a so-called type IV space-time interaction, assuming that for the i th area the parameter vector has an autoregressive structure on the time component and that each time point there is spatial correlation. The combination of a type IV space-time interaction, whereby time was modelled using a first-order random walk was chosen based on the WAIC, having a better model fit than other types of interactions and/or other ways to model time (WAIC difference ≥ 30).

While survey participants were more likely to be vaccinated than expected based on the national administrative data on vaccination uptake, adding an indicator for vaccination status (yes/no) interacting with time in weeks to a model that already accounted for all other variables improved model fit (WAIC difference of 964), but this had only small effects on post-stratified estimated levels of PCR positivity (Fig. 2)."

& re antibody models:

"As observed when focusing on swab positivity, the best fitting model was a model with a type IV space-time interaction and first-order random walk for time modelled in weeks (WAIC difference of ≥ 29). In contrast to the MRP model for swabs, an interaction between sex and time improved the model fit, meaning that all considered covariate terms were included in the final model."

Discussion:

"To obtain our estimates of PCR and antibody positivity over time and space, we extended an existing spatial regression and post-stratification approach by adding a time and space-time component. This new approach has the benefits of being more efficient than fitting separate models to each week, it enabled us to account for temporal correlation in the panel data and to better quantify variation and uncertainty in (local) trends."

2. Since the UK's CIS depends on voluntary participation, the manuscript emphasizes that vaccination status is an important confounder to be adjusted for in terms of selection bias. What about COVID-related symptoms? Will individuals have symptoms have higher tendency to participate the CIS than those without symptoms? If so, symptom status should also be accounted for.

Response: We do not think this comment would improve the analyses, as one should never try to adjust for a factor that is a consequence of the outcome, i.e. without infection no COVID-related symptoms. Given that the majority of people do not

experience infection at all during the early parts of the pandemic when participants have signed up – COVID-related symptoms cannot and should not be included as a covariate in an MRP model. Instead one could adjust for other factors, such as age, ethnicity, sex, and area, that are known/well-defined for all participants and drive potential acceptance of the invitation to participate in the survey.

Furthermore, as we have now further emphasises in the text, survey participants are followed longitudinally and tested using a fixed schedule.

3. Area-specific covariates, such as levels of deprivation and rurality percentages, are used for exploratory analyses after model estimation. Are they predictive covariates in the model? In the Discussion, the manuscript says they are not included in the model to avoid duplicated use. However, since the goal is to obtain stabilized spatial estimates, spatial covariates would play an important role in the model to avoid over-smoothing.

Response:

Ideally one would have household-level information on deprivation and rurality, however this is not available. While small area estimation methods sometimes indeed use additional contextual information to produce robust estimates of under- or unobserved geographic units. However, given that the sampling design of the CIS with 116 CIS area being designed to ensure a large number of participants in each area. As such there are no under- or unobserved geographic unit and contextual covariates are not expected to be add much here.

Therefore, we did, as explained in the discussion, use deprivation and rurality only for exploratory analyses after model estimation, to enable us to answer the question whether and why certain areas may have consistently higher PCR positivity than other areas.

4. In the Section of Statistical analysis, it writes "Post-stratification tables were based on the conditional distribution of age and sex by area from ONS. The conditional distribution of ethnicity by these categories were obtained from the ETHPOP database.^{5,27} Conditional distribution of vaccination uptake were obtained from the NIMS administrative data on vaccination uptake." This means that there is no existing joint distribution of age, sex, area, ethnicity, and vaccination status in the "population". How were these separate conditional distributions merged into a joint table? What are the assumptions? Will the uncertainty be accounted for?

Response:

We have modified this section now to explain in more detail what was done and assumed regarding the post-stratification tables:

"Distribution of characteristics in the target population

The Office for National Statistics provided weekly estimates of the proportion of individuals in each subgroup of the population based on NIMS data linked to Census 2011 data. Therefore, only participants that were already living in England during the Census 2011 were used to inform conditional estimates of vaccine uptake, thereby implicitly assuming that vaccination uptake in each subgroup of the population is the same for individuals from that subgroup living in England as those from the same subgroup of the population that moved to the England more recently. For children too

young to be young to be present in the Census 2011 we assumed that no vaccination under the age of 5 and half the coverage of that observed among 9-11 years old children – who could be present in Census 2011 – for those aged 5-8 years of age. Different assumptions, such as assuming no vaccinations under the age of 9 did not materially affect vaccine uptake estimates given the fact that vaccination uptake among children aged 9-11 years of age, the ages from which we extrapolated to younger ages were very rarely vaccinated during the study period.

However, by linking the NIMS data to Census 2011 data at an individual level we could estimate vaccination uptake over time by all variables considered here, including age, sex, ethnicity, and area. Estimates are available on a daily basis, but aggregated by week in line with the grouping performed for the survey data. To be able to estimate the number of people in each so-called post-stratification cell, including those that were not present in Census 2011, e.g. the number of vaccinated 35-49 year old males of black ethnicity living in a private household in Northumberland in the 20th week of the study period, we applied these conditional estimates of vaccination uptake percentage to the estimated number of individuals living in private households in 2020 for the corresponding categories (age, sex, ethnicity, area). The latter estimates focused on individuals living in private households to line up with the sampling base of the CIS. These estimates were generated by ONS by updating the 2011 census using the cohort component method, aging the population by 1 year each year, and incorporating births, deaths, immigration, emigration and people entering and leaving ‘special populations’ such as individuals in prisons. The conditional distribution of ethnicity by these categories were obtained from the ETHPOP database. Given that the ONS estimates do not come with a measure of uncertainty, uncertainty in the final post-stratifications table were not taken into account.”

Reviewer #4 (Remarks to the Author):

Summary

This manuscript adds to evidence that accurate, representative, small-area estimates of the prevalence of COVID-19 are unusual and hard to produce. The key issue is selective participation (or non-participation) that is related to the outcome(s) of interest - COVID-19 status of some sort. Prior work has revealed that selective non-response to erstwhile population-representative surveys can introduce important bias in estimated prevalence - e.g. political ideology associated with behaviors that affect COVID-19 status (PNAS, <https://doi.org/10.1073/pnas.202394711>). This work uses the same approach to evaluate the effect of not accounting for vaccination status when estimating infection prevalence and antibody levels using high quality, longitudinal data. The results demonstrate that accounting for vaccination status is essential, but that the effect is nuanced as the epidemic progresses and vaccination campaigns are increasingly

successful.

Data Methods

The data are unusually useful - high quality and sufficient quantity. With respect to methods, the results further validate an approach that involves model-based estimation using Bayesian hierarchical space-time models coupled with post-stratification to produce population-level estimates. Over the past few years this combination of methods has been shown to be generally superior in comparison to traditional frequentist estimation approaches, more robust, and the only viable approach when data are defective in various ways.

COVID-19 Status Estimates

With respect to producing large and small-area estimates of COVID-19 status, this work clearly demonstrates that including vaccination status in the models is essential; potentially large, consequential errors are possible when vaccination status is ignored.

Presentation and Reproducibility

1. The work is presented well in text and figures and there is a copy of the R script that was used for analysis. However, I could not find the statistical models or their priors written out anywhere in mathematical notation. This makes it hard or essentially impossible to completely interrogate or understand exactly what was done (or, it requires poring over the code very carefully, which I did not do). I strongly suggest that the statistical models and associated material (e.g. priors) be written out clearly using standard notation.

Response:

We have now more clearly written which covariates and interactions were included in the models and also have written out the models and priors and provided references that further explain constraints imposed for model identifiability.

2. As far as I can tell, the main challenge with this work is the 2011 baseline census to determine the size of each cell. I wonder if it would be possible to conduct a sensitivity analysis of some sort to gauge exactly how consequential this issue may be - a decade of population change could change things significantly.

Response:

We have now clarified how we used the baseline 2011 census, i.e. we do NOT assume that populations have not changed. In addition, we clarified our assumptions and the negligible impact of different assumptions for the age-group that was not present during the 2011 census:

“Distribution of characteristics in the target population

The Office for National Statistics provided weekly estimates of the proportion of individuals in each subgroup of the population based on NIMS data linked to Census 2011 data. Therefore, only participants that were already living in England during the Census 2011 were used to inform conditional estimates of vaccine uptake, thereby

implicitly assuming that vaccination uptake in each subgroup of the population is the same for individuals from that subgroup living in England as those from the same subgroup of the population that moved to the England more recently. For children too young to be present in the Census 2011 we assumed that no vaccination under the age of 5 and half the coverage of that observed among 9-11 years old children – who could be present in Census 2011 – for those aged 5-8 years of age. Different assumptions, such as assuming no vaccinations under the age of 9 did not materially affect vaccine uptake estimates given the fact that vaccination uptake among children aged 9-11 years of age, the ages from which we extrapolated to younger ages were very rarely vaccinated during the study period.

However, by linking the NIMS data to Census 2011 data at an individual level we could estimate vaccination uptake over time by all variables considered here, including age, sex, ethnicity, and area. Estimates are available on a daily basis, but aggregated by week in line with the grouping performed for the survey data. To be able to estimate the number of people in each so-called post-stratification cell, including those that were not present in Census 2011, e.g. the number of vaccinated 35-49 year old males of black ethnicity living in a private household in Northumberland in the 20th week of the study period, we applied these conditional estimates of vaccination uptake percentage to the estimated number of individuals living in private households in 2020 for the corresponding categories (age, sex, ethnicity, area). The latter estimates focused on individuals living in private households to line up with the sampling base of the CIS. These estimates were generated by ONS by updating the 2011 census using the cohort component method, aging the population by 1 year each year, and incorporating births, deaths, immigration, emigration and people entering and leaving 'special populations' such as individuals in prisons.

The conditional distribution of ethnicity by these categories were obtained from the ETHPOP database.^{5,27} Given that the ONS estimates do not come with a measure of uncertainty, uncertainty in the final post-stratifications table were not taken into account."

3. You may want to reference "*Estimating seroprevalence of SARS-CoV-2 in Ohio: A Bayesian multilevel poststratification approach with multiple diagnostic tests*" (<https://doi.org/10.1073/pnas.202394711>) that applied the same method to a similar situation with the added complexity of multiple tests but without the temporal dimension.

Response: We have added this reference.

REVIEWER COMMENTS

Reviewer #1 (Remarks to the Author):

I appreciate the authors extensive work I. Responding to my and other reviewer's comments. My comments are all satisfyingly addressed and I support acceptance of this manuscript.

Reviewer #3 (Remarks to the Author):

I appreciate the responses. However, the response to the question of including area-specific covariates is conflicting. The text writes:

"many areas had weeks with 2 or less positive samples in total despite the size of the survey (25th percentile of the number of positive tests per week was 2), emphasising the need for small-area estimation and potentially explaining why a better model fit was obtained for models with a type IV interaction and a random walk structure for time."

Hence we need a small-area estimate model that can stabilize estimates for small areas and also avoid over-shrinkage to the overall mean. But then in the response letter, it writes:

"While small area estimation methods sometimes indeed use additional contextual information to produce robust estimates of under- or unobserved geographic units. However, given that the sampling design of the CIS with 116 CIS area being designed to ensure a large number of participants in each area. As such there are no under- or unobserved geographic unit and contextual covariates are not expected to be add much here."

It is unclear about the benefit of the hierarchical model fitting. Does it improve the estimates in small areas? Does it cause overshrinking?

Reviewer #4 (Remarks to the Author):

I had two important suggestions for this manuscript: 1) provide a clear and detailed description of the methods, and 2) describe the base population used in the post-stratification step and explain how it is fit for purpose. You essentially chose to ignore (1) and provided a good response for (2).

Please provide a supplemental file that describes the data, all modifications that were made to the data in detail, models, estimation methods, validation work, and final results using equations and explaining your choices along the way. Two examples:

1.

<https://journals.plos.org/plosone/article/file?type=supplementary&id=10.1371/journal.pone.0210645.s001>

2. https://www.ncbi.nlm.nih.gov/pmc/articles/PMC5154628/bin/NIHMS814893-supplement-Supplementary_Material.pdf

I have read your R code. Please improve the documentation and link/cross reference the code to the text and equations that you provide in the new supplemental material.

Reviewer #4 (Remarks on code availability):

See above.

Response to the reviewers:

REVIEWER COMMENTS

Reviewer #1 (Remarks to the Author):

I appreciate the authors extensive work I. Responding to my and other reviewer's comments. My comments are all satisfyingly addressed and I support acceptance of this manuscript.

Response: Thank you for the positive comments.

Reviewer #3 (Remarks to the Author):

I appreciate the responses. However, the response to the question of including area-specific covariates is conflicting. The text writes:

"many areas had weeks with 2 or less positive samples in total despite the size of the survey (25th percentile of the number of positive tests per week was 2), emphasising the need for small-area estimation and potentially explaining why a better model fit was obtained for models with a type IV interaction and a random walk structure for time."

Hence we need a small-area estimate model that can stabilize estimates for small areas and also avoid over-shrinkage to the overall mean. But then in the response letter, it writes:

"While small area estimation methods sometimes indeed use additional contextual information to produce robust estimates of under- or unobserved geographic units. However, given that the sampling design of the CIS with 116 CIS area being designed to ensure a large number of participants in each area. As such there are no under- or unobserved geographic unit and contextual covariates are not expected to be add much here."

It is unclear about the benefit of the hierarchical model fitting. Does it improve the estimates in small areas? Does it cause overshrinking?

Response: Thank you for highlighting this, and allowing us to clarify this point. While we have a large number of participants tested, due to the low number of infected individuals, i.e. low number of positive samples, small area-estimation techniques could still be potentially relevant when focusing on PCR positivity. We have now evaluated whether a model specifically including contextual information, in this case the CIS area-specific levels of urbanicity and deprivation, as explanatory variables in the MRP model would improve model fit. In the previous iteration of the manuscript, these two variables were shown to explain a lot of the variation between CIS-area specific estimates based on an analysis relating these variables measured at CIS-area level with post-stratified estimates from an MRP model without those covariates. However, this was actually associated with a small increase in the WAIC, and hence worse model fit, compared to the existing model with contextual information only captured through the effects for area, time and the type-IV interaction (not explicitly including deprivation and urbanicity). We have added this information to the main text of the manuscript, making clear that the

considered context-level covariates did not lead to an improvement in model fit and based on the WAIC adding the considered context variables seems to lead to overfitting.

It is only possibly quantify the exact benefit from hierarchical model fitting with certainty in simulations where the ground truth is known. However, numerous simulation studies have already shown the benefits of hierarchical model fitting for obtaining representative estimates at the (sub)national level, such as the study from Goa et al already referenced. Given this, and the amount of material already presented, we feel that another extensive simulation study would be a paper on its own and therefore out of scope here. This is particularly the case as the comparison with the raw data shows that using the MRP hierarchical model indeed moves estimates in the expected direction, i.e. higher swab positivity due to vaccinated individuals and white individuals being slightly over-represented in the survey.

In addition, we further clarified that due to low positivity rates a low number of individuals tested positive despite large sample size:

"because of low positivity rates, many areas had weeks with 2 or fewer positive samples in total despite the size of the survey meaning several thousand were tested (25th percentile of the number of positive tests per week was 2), emphasising the need for small-area estimation and potentially explaining why a better model fit was obtained for models with a type IV interaction and a random walk structure for time

Reviewer #4 (Remarks to the Author):

I had two important suggestions for this manuscript: 1) provide a clear and detailed description of the methods, and 2) describe the base population used in the post-stratification step and explain how it is fit for purpose. You essentially chose to ignore (1) and provided a good response for (2).

Please provide a supplemental file that describes the data, all modifications that were made to the data in detail, models, estimation methods, validation work, and final results using equations and explaining your choices along the way. Two examples:

1. <https://journals.plos.org/plosone/article/file?type=supplementary&id=10.1371/journal.pone.0210645.s001>
2. https://www.ncbi.nlm.nih.gov/pmc/articles/PMC5154628/bin/NIHMS814893-supplement-Supplementary_Material.pdf

I have read your R code. Please improve the documentation and link/cross reference the code to the text and equations that you provide in the new supplemental material.

Response:

We do not agree with the reviewer's comment suggesting that we did not provide further information on the methods as they had requested. In the previous revision, we had already added additional details into the Methods section about the model and priors required to recreate the model, specifically

The best fitting model was a model with a first-order random-walk for time (in weeks) and a type-IV interaction between CIS area and time, i.e. a model where the temporal dependency structure for each area also depends on the temporal pattern of the neighbouring areas. The linear predictor of this spatiotemporal model accounts additively for the temporal effects, spatial effects, and the spatiotemporal interactions, and is linked to the expected value of the response y through a logit link function, such that $E(y) = \text{logit}^{-1}(\eta)$.

For the final model with a first-order random walk for time and a type IV interaction between time and CIS area, we used penalised complexity (PC) priors - PC-prior(1,0.1) – for the precision of the random effects of covariates with multiple categories (ethnicity and age), the first-order random walk for time, and the space-time interaction. The precision of the ethnicity*time, age*time, and vaccination*time interactions was modelled using a PC-prior(1,0.01). For the other fixed effects (vaccinated [yes/no] and sex) we used priors with a mean of zero and precision of 0.01 (standard deviation of 10). Based on Gao et al – who found that the default BYM2 hyperprior specification for the mixing parameter ϕ in INLA performed well in situations with and without a true spatial structure,⁵ we used the default PC-prior(0.5,0.5) for the mixing parameter of the BYM2 spatial prior.”

In addition, we had also added into the Methods further information on how the space time type IV interaction was implemented:

“... a type IV space-time interaction assumes that the temporal dependency structure for each area also depends on the temporal pattern of the neighbouring areas.³² Sum-to-zero constraints appropriate for the type of interaction and the type of random walk (first- or second-order) used to model time were imposed to ensure identifiability of the model in line with Goicoa et al and the bigDM R package (see also <https://emi-sstcdapp.unavarra.es/bigDM/bigDM-3-fitting-spatio-temporal-models.html>).^{34,35} “

Therefore, and particularly reviewing Methods and supplementary material from other recent papers published in Nature Communications, we do feel that this comment may be partly driven by style preference. We would be happy to follow editorial guidance on this.

However, to make it easier to understand the code, we have now written out the sum-to-zero constraint of the final model (while keeping a reference to Goicoa paper describing the constraints for other types of combinations), and annotated the R code further with reference to the manuscript and relevant other sources (for the latter see https://github.com/pouwelskb/spatiotemporal_mrp_covid/).

Specifically we added the following additional information to the main-text:

“Goicoa et al explain in detail which identifiability constraints are necessary dependent on how time is modelled and the type of interaction between space and time.³⁴ The appendix of that paper list constraints required for different combinations of space-time interactions (I-IV) combined with first-order or second-order random walks for time.³⁴ Given that the best fitting model was a first-order random-walk for time and a type-IV interaction between CIS area and time, below identifiability constraints were imposed.³⁴”

$$\sum_{t=1}^T \delta_{it} = 0, \text{ for } i = 1, \dots, S$$

$$\sum_{i=0}^S \xi_i = 0, \quad \sum_{t=1}^T \gamma_t = 0$$

$$\sum_{i=1}^S \delta_{it} = 0, \text{ for } t = 1, \dots, T$$

where ξ represents the spatial random effect, γ the temporal random effect (first-order random walk), and δ represents the structured interaction random effect (type IV interaction)."